# The *Torreya grandis* genome illuminates the origin and evolution of gymnosperm-specific sciadonic acid biosynthesis

Heqiang Lou [1,7], Lili Song[1,7], Xiaolong Li[2,3,7], Hailing Zi[4], Weijie Chen[1], Yadi Gao[1], Shan Zheng[1], Zhangjun Fei [5,6] ✉, Xuepeng Sun [2,3] ✉ & Jiasheng Wu [1] ✉

*Torreya* plants produce dry fruits with assorted functions. Here, we report the 19-Gb chromosome-level genome assembly of *T. grandis*. The genome is shaped by ancient whole-genome duplications and recurrent LTR retrotransposon bursts. Comparative genomic analyses reveal key genes involved in reproductive organ development, cell wall biosynthesis and seed storage. Two genes encoding a $C_{18}$ $\Delta^9$-elongase and a $C_{20}$ $\Delta^5$-desaturase are identified to be responsible for sciadonic acid biosynthesis and both are present in diverse plant lineages except angiosperms. We demonstrate that the histidine-rich boxes of the $\Delta^5$-desaturase are crucial for its catalytic activity. Methylome analysis reveals that methylation valleys of the *T. grandis* seed genome harbor genes associated with important seed activities, including cell wall and lipid biosynthesis. Moreover, seed development is accompanied by DNA methylation changes that possibly fuel energy production. This study provides important genomic resources and elucidates the evolutionary mechanism of sciadonic acid biosynthesis in land plants.

The emergence of seed plants consisting of angiosperms and gymnosperms marked a momentous event in the evolution of land plants and the change of earth environments. Angiosperms and gymnosperms diverged in the Lower Mississippian[1], followed by rapid radiation of flowering plants resulting in approximately 352,000 extant species on Earth compared to only 1000 species of gymnosperms. There is a variety of morphological/anatomical diversity and metabolic versatility between angiosperms and gymnosperms, but the underlying genetic and biochemical mechanisms are largely elusive[2,3].

*Torreya grandis*, a gymnosperm species belonging to a small genus of the yew family (Taxaceae), is a useful multipurpose tree, providing timber, medicine, edible seeds and oil[4] (Fig. 1a). The first credible record of *T. grandis* as a medicinal source appears in Classic of

the Materia Medica during the Three Kingdoms of China and dates back to the beginning of the 3rd century AD[5]. *T. grandis* is the only species in Taxaceae with edible seeds, which have been used as food for thousands of years in China due to their unique flavor and beneficial components[5,6]. Oils are enriched in seeds of *T. grandis* with an average content of 45.80−53.16%[7]. Sciadonic acid (SCA), a non-methylene-interrupted ω6 fatty acid, has been found as one of the major components in fatty acid composition of the kernel oil[7]. SCA has positive effects on human health, and functions in reducing inflammation, lowering triglycerides, preventing blood clots and regulating lipid metabolism[8–10]. Production of SCA has been detected in different lineages of gymnosperms and a handful of algae and ferns[11]. However, SCA is generally absent in flowering plants, with the exception of a few

[1]State Key Laboratory of Subtropical Silviculture, Zhejiang A&F University, Hangzhou 311300 Zhejiang, China. [2]Collaborative Innovation Center for Efficient and Green Production of Agriculture in Mountainous Areas of Zhejiang Province, Zhejiang A&F University, Hangzhou 311300 Zhejiang, China. [3]Key Laboratory of Quality and Safety Control for Subtropical Fruit and Vegetable, Ministry of Agriculture and Rural Affairs, Hangzhou 311300 Zhejiang, China. [4]Novogene Bioinformatics Institute, 100083 Beijing, China. [5]Boyce Thompson Institute, Cornell University, Ithaca, NY 14853, USA. [6]U.S. Department of Agriculture-Agricultural Research Service, Robert W. Holley Center for Agriculture and Health, Ithaca, NY 14853, USA. [7]These authors contributed equally: Heqiang Lou, Lili Song, Xiaolong Li. ✉e-mail: zf25@cornell.edu; xs57@zafu.edu.cn; wujs@zafu.edu.cn

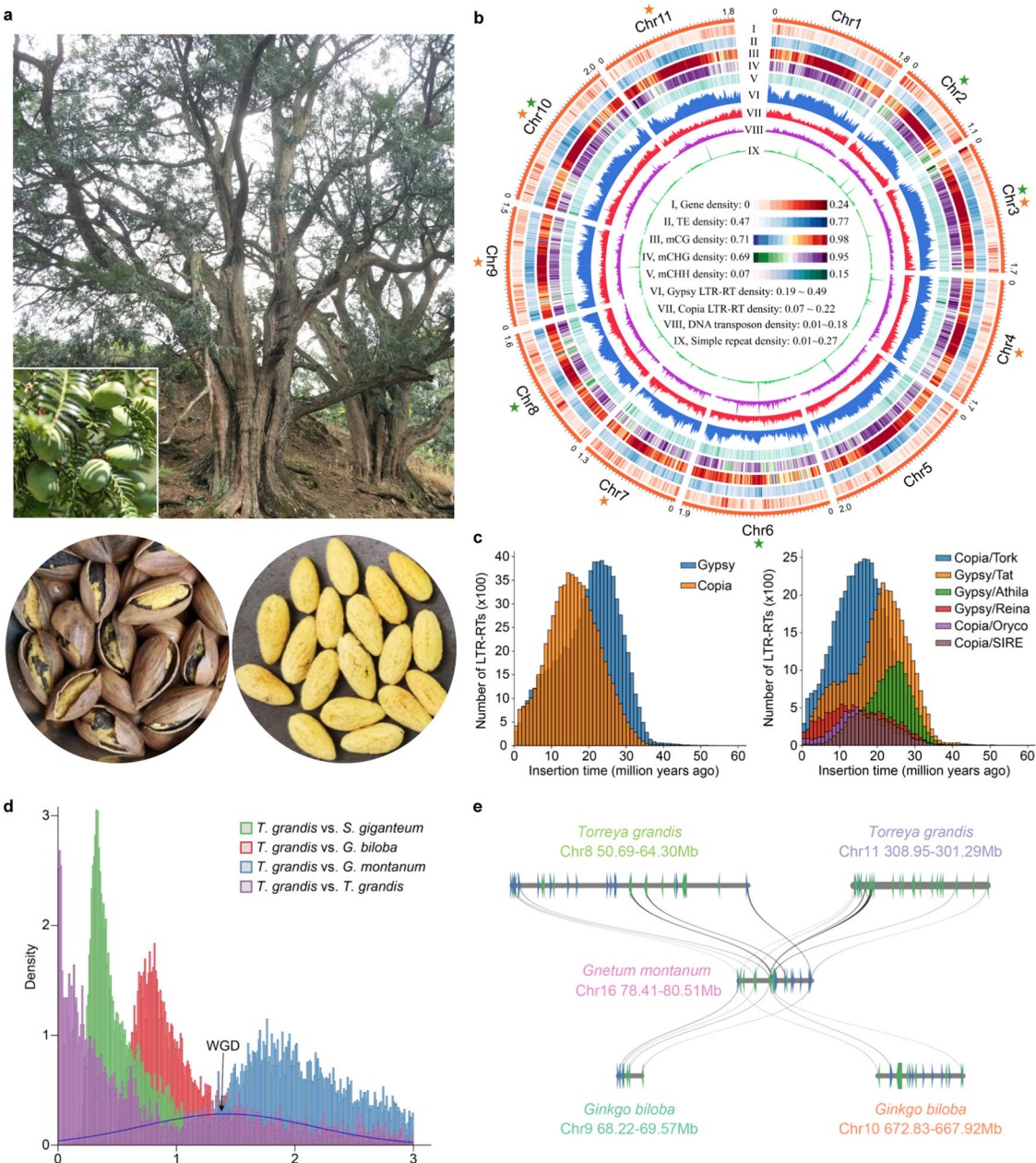

**Fig. 1 | Genome of *T. grandis*. a** Tree and fruit set of *T. grandis*. The lower panel shows the processed dry seed and its edible part (endosperm). **b** Circos plot of *T. grandis* genome and genomic features encoded by the chromosomes. Each feature was calculated based on a 10-Mb window across the chromosomes. Colored stars indicate the presence of telomeric sequences on 5′- (green) or 3′-end (orange) of the chromosome. **c** Distribution of LTR-RT insertion time. The left panel shows all members of *Gypsy* and *Copia* families and the top six most abundant subfamilies are shown on the right panel. **d** *Ks* distribution of orthologues among *T. grandis*, *Sequoiadendron giganteum*, *Ginkgo biloba* and *Gnetum montanum*. *Ks* of paralogues in *T. grandis* was fitted with Guassian mixture model and the putative ancient WGD is indicated. **e** Micro-collinearity between genomes of *T. grandis*, *G. biloba* and *G. montanum*. Source data are provided as a Source Data file.

lower eudicots (e.g., Ranunculaceae)[12], thus leaving a puzzle on its origin and evolution in green plants.

Genome sequences are key to addressing critical questions of plant evolution. Analyses of genomes of representative gymnosperms have shown unique aspects of gene and genome evolution that distinguish themselves from flowering plants[13–19]. However, understanding the biological and evolutionary mechanisms of phenotypic diversity between flowering and non-flowering plants is still challenging, partly due to the limited availability of genomic resources of gymnosperms.

In this study, we assemble a chromosome-scale reference genome for *T. grandis*, accompanied by transcriptome and methylome

profiling in multiple tissues. Our data, through comparative genomic analyses, unravel interesting footprints associated with morphological diversity of major land plant lineages, and discover and validate two key enzymes that are responsible for SCA biosynthesis. Information provided by this work will be useful for strategic design on improvement of SCA production, and promote the utilization of *Torreya* genetic resources.

## Results

### Genome assembly and annotation

We generated a total of 1.93 Tb of Illumina and 463.7 Gb of PacBio HiFi reads for *T. grandis* (Supplementary Data 1), representing 96.5× and

23.2× coverage, respectively, of the *T. grandis* genome that had an estimated size of ~20 Gb according to the *k*-mer analysis of the Illumina reads (Supplementary Fig. 1). The final assembly had a size of 19,050,820,213 bp, comprising 11,811 contigs with an N50 size of 2.82 Mb (Supplementary Table 1). Using Hi-C reads of approximately 106.2× coverage, 18.87 Gb (99.1%) of the assembled contigs were grouped into 11 chromosomes (Fig. 1b and Supplementary Fig. 2). All 11 chromosomes were found to be enriched with the 101-bp repetitive sequence unit that resembles the tandem centromeric satellite repeat known as the landmark of centromeres, while 9 chromosomes harbored telomeric sequences (5′-TTTAGGG-3′) in at least one end (Fig. 1b). Assessment of the *T. grandis* genome using Merqury[20] revealed a consensus quality score of 46.9, equivalent to a base accuracy of 99.998%. BUSCO[21] evaluation indicated that 1386 out of 1614 land plant conserved orthologues were successfully captured by the *T. grandis* assembly, which was comparable to that of other gymnosperm genome assemblies (Supplementary Table 2). The LTR assembly index (LAI) for the *T. grandis* genome was 10.7, which was higher than the proposed standard for a reference genome[22]. These together with the high DNA (99.48%) and RNA (up to 97.5%) read mapping rates suggested the high quality of the *T. grandis* genome assembly.

The *T. grandis* genome assembly harbored 11.4 Gb (59.8%) of repetitive sequences, of which LTR retrotransposons (LTR-RTs; 87.0%) were predominant, followed by DNA transposons (7.1%) and long interspersed nuclear elements (LINEs; 3.1%) (Supplementary Data 2). The proportion of *Copia* LTR-RTs (11.6%) was relatively higher in *T. grandis* than in other gymnosperms, possibly due to recent species-specific bursts occurring in multiple subfamilies of LTR-RTs (Fig. 1c). Most of the LTR-RT expansions in gymnosperms took place between 25-7 million years ago (mya; Supplementary Fig. 3a), overlapping with the geological time of Miocene epoch (23.03–5.33 mya) when the earth cooled down towards ice ages[23], suggesting a potential environmental effect on the genome size evolution of gymnosperms.

A total of 47,089 protein-coding genes were predicted in the *T. grandis* genome, of which 46,338 were supported by homology and/or transcriptome evidence (Supplementary Table 1). Intron size is more variable in gymnosperms than in angiosperms (Supplementary Fig. 3b), which is attributed to the expansion of LTR-RTs. In plants, LTR-RTs can be eliminated through unequal recombination, creating solo-LTRs in the genome. The solo:intact LTR ratio is high in *T. grandis* (4.3) and other gymnosperms including *Taxus wallichiana* (5.5)[18], *Ginkgo biloba* (4.26), *Welwitschia mirabilis* (3.87), and *Gnetum montanum* (2.07)[16]. Since gymnosperm genomes are enriched with ancient LTR-RTs (10–30 mya)[14–19], we hypothesize that elimination of ancient LTR-RTs without recent expansion may have contributed to their high solo:intact LTR ratios. This is in contrast with the small-genome angiosperms, in which LTR-RT bursts are more recent (<4 mya)[24]. Epigenetic silencing of transposons and pericentromeric repeats is mediated by RNA-directed DNA methylation (RdDM) and 24-nt hetsiRNAs[25]. The *T. grandis* genome encoded homologs of key components of the RdDM pathway (Supplementary Data 3); however, small RNA profiling of seven tissues showed that 21-nt sRNAs were the most abundant in *T. grandis*, contrasting to the most abundant 24-nt sRNAs in angiosperms, while productions of 22-nt and 24-nt sRNAs were tissue-specific (Supplementary Fig. 4). This pattern is similar to that found in conifers[13,26] and *Welwitschia mirabilis*[16]. Nonetheless, further extensive sampling from additional tissues and stages would provide deeper insights into the divergence of sRNA processing between gymnosperms and angiosperms.

## Ancient whole-genome duplication

Whole-genome duplications (WGDs) have occurred across the breadth of eukaryote phylogeny[27]. In gymnosperms, several WGDs have been recognized although some of them remain in controversy[16–19,28]. The *Ks*

distribution of 3859 paralogue groups within *T. grandis* indicated the absence of recent WGDs. However, we observed a peak of *Ks* ranging from 1 to 2 and a summit at 1.4, representing a potential ancient WGD that occurred in the common ancestor of conifers and ginkgophytes, a lineage diverged from gnetophytes (Fig. 1d). We then used a tree-based approach[29], which calculates the frequency of gene duplication on every branch of a phylogeny by reconciliation of gene tree and species tree, to cross validate the WGD event. Analysis of 19,649 gene trees from eight selected species led to the discovery of three ancient WGD signals including two (zeta and omega) reported previously[17,28] and one that was consistent with the *Ks* analysis (Supplementary Fig. 5). Whole-genome comparison showed high collinearity among genomes of *T. grandis* and two evolutionarily distant gymnosperms, *Sequoiadendron giganteum* and *Ginkgo biloba* (Supplementary Fig. 6), and also revealed traces of collinear blocks that were duplicated in both *T. grandis* and *G. biloba* but not in *Gnetum montanum*, agreeing with the timing at which the newly discovered WGD occurred (Fig. 1e).

## Gene family evolution

We identified 19,362 orthologous groups (gene families) in 19 plant species comprising 7 gymnosperms and 12 representative species in major green plant lineages. Phylogeny and molecular dating using 219 low-copy gene families indicated that *T. grandis* separated from *T. wallichiana* around 68.5 mya (Fig. 2a). Expansion of gene families has been implicated in tight associations with morphological innovations[30,31]. Through reconstruction of gene family evolution, we found that bursts of gene family expansions coincided with major transitions of plant adaptation (Fig. 2a). Massive gene family expansion ($n = 417$, $P < 0.05$) was observed in the common ancestor of land plants, and subsequently in the extinct ancestors leading to seed plants ($n = 575$), angiosperms ($n = 432$) and various lineages of gymnosperms ($n = 428–818$). Functions of expanded gene families were mostly associated with plant organ development, response to biotic (e.g., bacteria and fungi) and abiotic (e.g., water deprivation, light, temperature, and salt) stresses, and biosynthesis and signaling of plant hormones (Supplementary Data 4). Many of the gene families were expanded continuously towards the evolution of higher plants, suggesting that gene duplication, possibly followed by sub/neo-functionalization, provides genetic foundation for morphological diversity and environmental adaptation in plants. Among the gene families that were significantly expanded in *T. grandis*, many of them encoded pfam domains associated with important biological functions, including lipid transfer (Oleosin and PF14368), biotic and abiotic stress responses (PF00201 and PF03018) and secondary metabolism (PF00067) (Supplementary Data 5). The *T. grandis* genome lacked the orthologues of taxadiene synthase, a core component of paclitaxel biosynthesis, explaining the absence of paclitaxel and relevant metabolites in this species.

## Reproductive organ genes

Gymnosperms have unenclosed or naked seeds on the surface of scales or leaves, while flowers and fruits are angiosperm innovations. Phylogeny-based homolog search using well-studied flower development genes[32] showed sporadic distribution of these homologs in gymnosperms and non-seed plants (Supplementary Data 6), indicating the stepwise emergence accompanied with secondary loss of flower development genes during the evolution of land plants, as exemplified by *NOP10* (required for female gametophyte formation in flowers)[33] and *WUS* (required for shoot and floral meristem integrity)[34] genes that emerged early in land plants and were subsequently lost in both *T. grandis* and *T. wallichiana* (Supplementary Data 6).

The MADS-box family genes are a class of transcription factors involved in the regulation of floral organ specificity, flowering time, and fruit development. We identified 23 MIKC$^C$ MADS-box genes in *T. grandis*, including homologs of genes in the ABCE model of floral

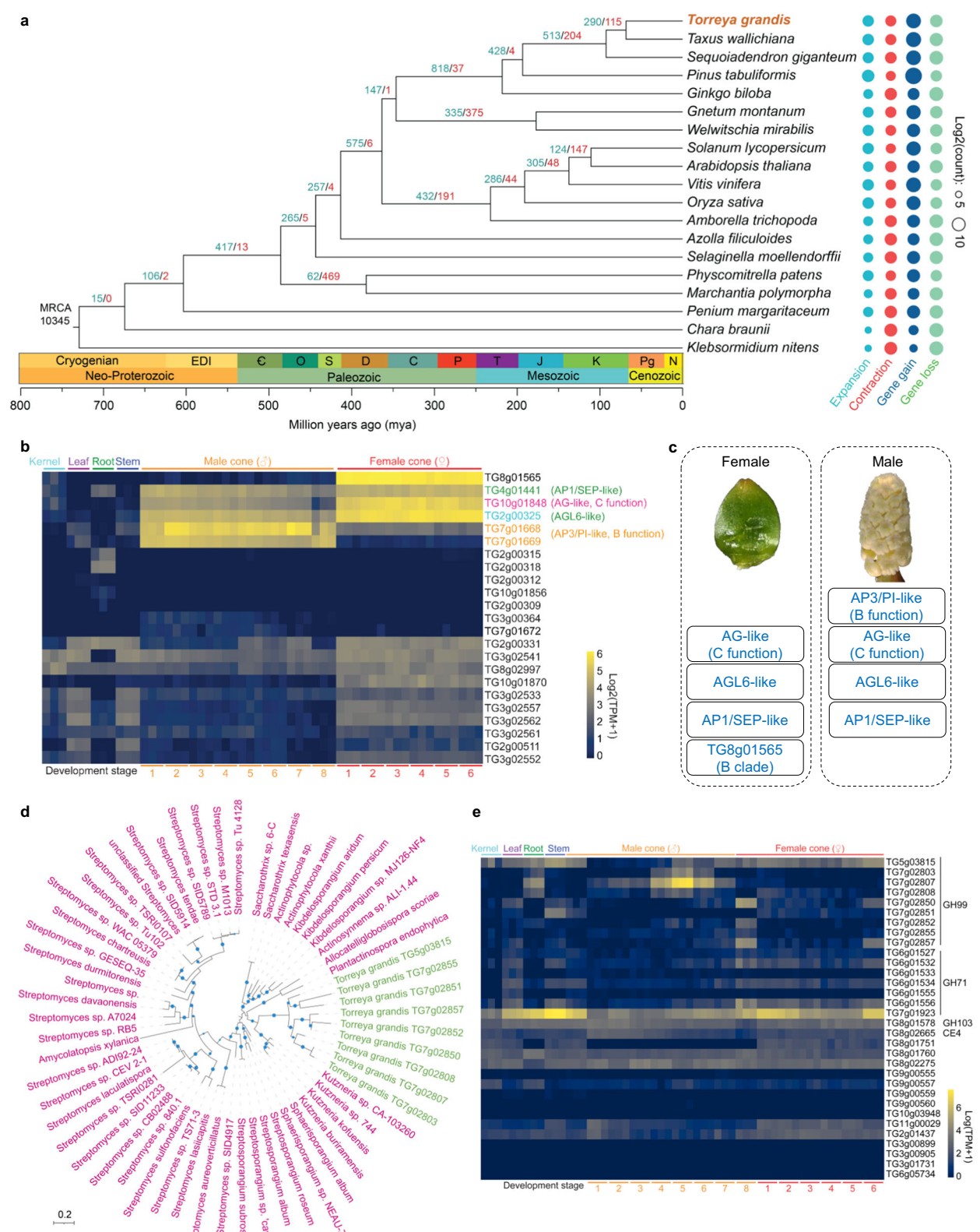

organ identity[35]. These included one AP1/SEP-like gene (A or E function), two AP3/PI-like genes (B function) and six AG-like genes (C function) (Supplementary Fig. 7). Transcriptome analysis of 18 samples from vegetative and reproductive organs revealed six MADS-box genes that were highly expressed in male and/or female cones of *T. grandis*, among which the two tandemly duplicated AP3/PI-like genes (*TG7g01668* and *TG7g01669*) were predominantly expressed

in the male cones, while one AG-like gene (*TG10g01848*) was expressed in male but upregulated by 6.6-fold in the female cones (Fig. 2b). Recent studies suggest that *AGL6*, member of an ancient subfamily of MADS-box genes, is involved in the E function of floral development in rice, maize and wheat[36,37], while participates in A function in the basal angiosperm *Nymphaea colorata*[38]. In *T. grandis*, the AGL6-like gene (*TG2g00325*) showed an expression pattern similar to that of the

**Fig. 2 | Gene family evolution in plants. a** Gene family expansion and contraction during the evolution of green plants. The maximum likelihood phylogeny was built with 219 low-copy orthologous groups. Gene family analysis was started with 10,345 orthologous groups that were shared by the most recent common ancestor (MRCA) of green plants. Numbers on branches are the sizes of expanded (blue) and contracted (red) gene families at each node. Colored pies on the right represent the sizes of expanded/contracted gene families as well as gained/lost genes for each leaf node of the tree. **b** Expression of MIKC$^C$ type MADS-box genes in vegetative and reproductive tissues of *T. grandis*. **c** Proposed reproductive organ identity genes in *T. grandis*. The AP3/PI-like genes (*TG7g01668* and *TG7g01669*) and *TG8g01565* were predominantly expressed in male and female cones, respectively. The AG-like (*TG10g01848*), AGL6-like (*TG2g00325*) and AP1/SEP-like (*TG4g01441*) genes were expressed in both female and male cones, with the first two showing a pattern biased to female cones. **d** Maximum likelihood phylogeny showing the bacterial origin of *T. grandis* genes. Blue pies indicate bootstrap support greater than 80% at the corresponding branches. **e** Expression of putative horizontally transferred genes in different tissues. Source data are provided as a Source Data file.

C function genes, whereas the *AP1/SEP*-like gene (*TG4g01441*) was expressed at a moderately high level in both male and female cones, resembling an ancestral role of E function. Interestingly, the most highly expressed MADS-box gene (*TG8g01565*) was exclusively activated in the female cones (Fig. 2b). This gene was phylogenetically clustered with B clade genes comprising *AP3*, *PI* and B-sister genes *TT16* and *GOA* (Supplementary Fig. 7); however, its expression pattern was opposite to that of the *AP3/PI*-like genes. In conclusion, our finding on the involvement of additional MADS-box genes in seed development of gymnosperms supports the basic "BC" model, where the C function genes are generally expressed in reproductive male and female organs and the B function genes are restricted to male reproductive organs[39], and suggests a more sophisticated regulatory system for reproductive organ development in gymnosperms (Fig. 2c).

### Seed storage proteins

The protein content of *T. grandis* seeds ranges from 10.34% to 16.43% depending on cultivars[7]. Genes encoding seed storage proteins (SSPs) including 2S albumins ($n = 0–7$), 7S globulins ($n = 1–9$) and 11S globulins ($n = 2–14$) were identified in *T. grandis* and other gymnosperms but not in the earlier forms of plants (Supplementary Data 7), suggesting their origin in seed plants. Transcriptome analysis showed that genes encoding 2S albumins and 7S globulins were expressed at an exceptionally high level (average transcripts per million (TPM) = 14,125) in the kernel of *T. grandis* seeds and the expression was increased during seed development (Supplementary Fig. 8a). In contrast, all SSP genes including 11S globulin genes, which were moderately expressed in the kernel, remained transcriptionally inactive in the vegetative tissues (Supplementary Fig. 8a). The 2S albumin proteins harbor numerous cysteine residues to form disulfide bridge within and between subunits[40]. We found that all these residues were conserved in *T. grandis* although the whole protein sequences were considerably divergent from the angiosperm counterparts (Supplementary Fig. 8b). Homology modeling revealed a high degree of protein structure conservation between the 2S albumin proteins from *T. grandis* (e.g., TG11g02972) and sunflower, particularly in the region where α-helices are formed (Supplementary Fig. 8c). Likewise, most of residues involved in trimer formation and stabilization as well as in correct globular folding of 11S globulins from flowering plants[41] were conserved in *T. grandis* (Supplementary Fig. 9). Overall, the gene expression and structural analyses suggest a conservative role of the major SSPs in both gymnosperms and angiosperms.

### Cell wall biosynthesis and horizontal gene transfer

Gymnosperms are mainly woody plants, and their genomes encode a large set of carbohydrate active enzymes (CAZymes) whose functions are closely associated with cell wall biosynthesis. Among the 19 selected representative plant species, *T. grandis* harbored more CAZymes than most others, particularly in the families of glycoside hydrolases (e.g., GH1, GH16, GH18, GH19, GH27, GH71, GH99, and GH152), GT61 glycosyltransferases, and PL1 polysaccharide lyases (Supplementary Data 7), many of which were also expanded in other gymnosperms. In contrast to most CAZyme families that were universally present in plants, we identified four families comprising 18 genes, GH71 ($n = 7$),

GH99 ($n = 9$), GH103 ($n = 1$), and CE4 ($n = 1$), that were present only in gymnosperms and prior lineages but not in angiosperms (Supplementary Data 8). Phylogenetic analysis showed that these families were of possible bacterial origin (Fig. 2d and Supplementary Fig. 10). Through systematic analysis, we identified 14 additional *T. grandis* genes that were derived from horizontal gene transfers (HGTs; Supplementary Table 3). Most of these genes were expressed in different tissues of the plant (Fig. 2e), reinforcing the contribution of HGTs in the evolution of land plants[42].

Lignin is a major component of plant secondary cell wall and is derived from p-hydroxyphenyl (H), guaiacyl (G), and syringyl (S) monolignols. S-lignin is restricted to flowering plants and some lycophytes, whereas G- and H-lignin are fundamental to all vascular plants[2]. Consistently, two key genes for S-lignin biosynthesis, *F5H* and *COMT*, were found only in angiosperms but not in gymnosperms. Unlike angiosperms in which vessels comprise major water-conducting elements in xylem[43], gymnosperm woods are mainly composed of tracheids[2]. Vessel differentiation is regulated by VASCULAR-RELATED MAC-DOMAIN (VND) proteins[44], while fiber development is associated with NAC SECONDARY WALL THICKENING PROMOTING FACTOR (NST)/SECONDARY WALL-ASSOCIATED NAC DOMAIN (SND) proteins[45]. The *T. grandis* genome encoded genes homologous to *VND4/5/6*, but lacked homologs of *VND1/2/3*, *NST* and *SND1* (Supplementary Fig. 11), which, combined with the finding of divergent regulatory networks of *VND/NST* homologs in conifer and flowering plants during wood formation[46], suggests a close link between vessel formation and the emergence of master NAC transcription factors as well as their regulatory networks in angiosperms.

### Discovery of key genes for sciadonic acid biosynthesis

Sciadonic acid (SCA) is a $\Delta^5$-olefinic fatty acid and its biosynthesis requires the activity of $C_{18}$ $\Delta^9$-elongase and $C_{20}$ $\Delta^5$-desaturase that uses 18:2-phosphatidylcholine (PC) as the initial substrate (Fig. 3a). $\Delta^5$-desaturases are known as the 'front-end' desaturases[47], which usually encode a cytochrome b5-like heme/steroid binding domain (PF00173) and a fatty acid desaturase domain (PF00487), whereas the $\Delta^9$-elongases encode a GNS1/SUR4 family domain (PF01151) for long chain fatty acid elongation. The *T. grandis* genome encoded four desaturase genes and four elongase genes based on the domain search. However, only one desaturase (TgDES1) showed high similarity with the previously reported $\Delta^5$-desaturase in *Anemone leveillei*[48], while two elongases were considered as putative $\Delta^9$-elongases but only one (*TgELO1*) was highly expressed in seed kernels (Supplementary Fig. 12). Since the unsaturated fatty acids are abundant components of seed oils, we investigated the expression of *TgDES1* and *TgELO1* during seed maturation. We found that SCA was accumulated in mature seeds, accompanied with the increased expression of *TgDES1*. Similar trend was observed for the expression of *TgELO1* and the content of its putative product cis-11,14-eicosadienoic acid (Fig. 3b, c). Study of subcellular localization showed that both TgELO1 and TgDES1 were co-localized with the marker of endoplasmic reticulum (ER) in *N. benthamiana* leaves (Fig. 3d), suggesting that they were bound to ER membrane, consistent with the subcellular location of known desaturases and elongases[49]. To further verify their function in SCA

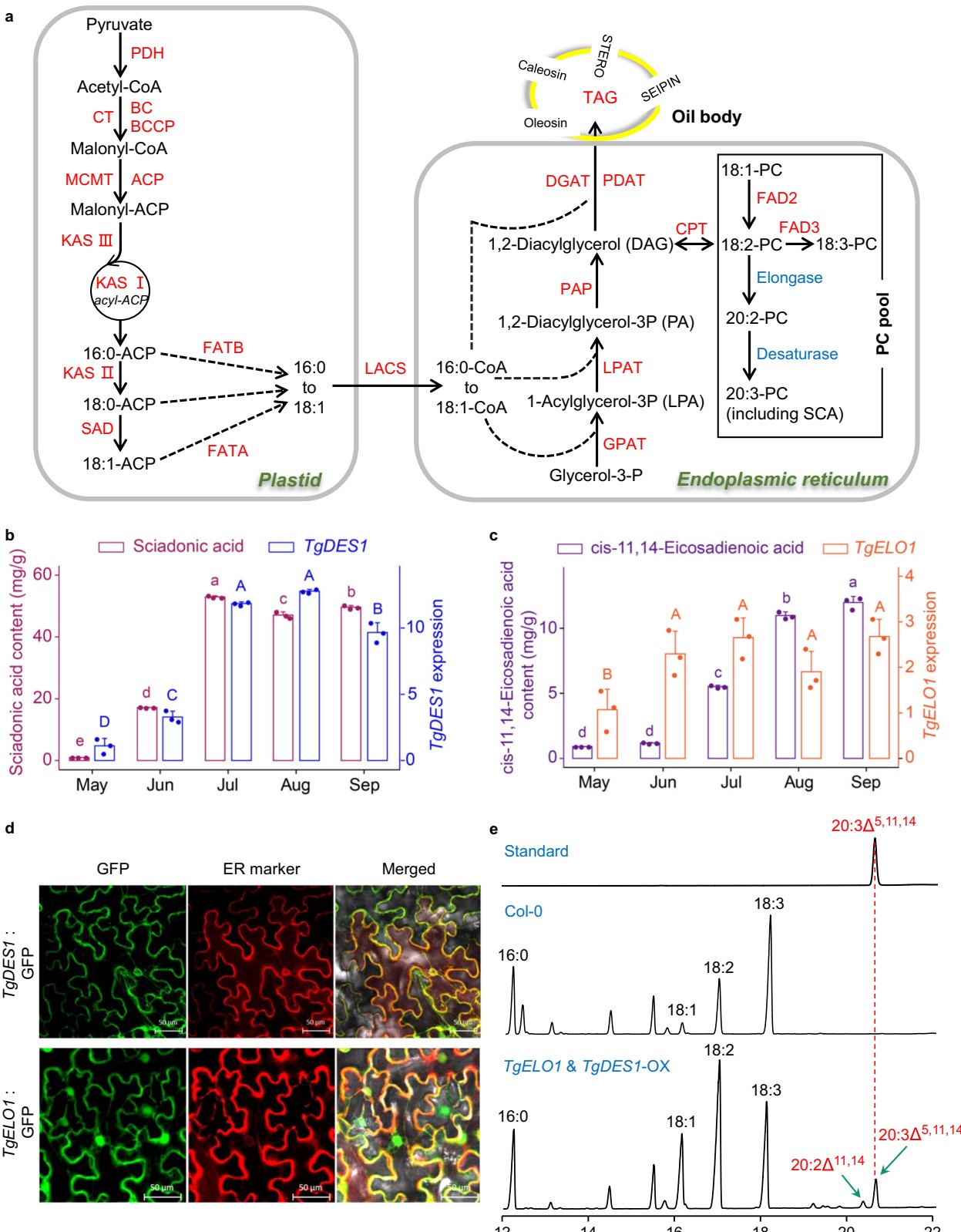

biosynthesis, we overexpressed both *TgELO1* and *TgDES1* in *A. thaliana*, which neither encodes orthologues of *TgELO1* and *TgDES1* nor produces SCA or its precursor 20:2$^{\Delta11,14}$-PC. Gas chromatography analysis showed that SCA was successfully synthesized in seeds of the transgenic line expressing *TgDES1* and *TgELO1*, demonstrating that *TgELO1* and *TgDES1* are capable of synthesizing SCA in *T. grandis* (Fig. 3e).

**Origin and evolution of plant Δ⁵-desaturases and Δ⁹-elongases**
Phylogenetic analysis of desaturases in green plants (Viridiplantae) showed that TgDES1 clustered with desaturases exclusively from non-angiosperm organisms, and this monophyletic clade was close to the family containing sphingolipid desaturases including AtSLDs from Arabidopsis (Fig. 4a). Interestingly, the TgDES1 clade clearly separated from the group harboring AL10 and AL21, two proteins that were found

**Fig. 3 | Characterization of genes responsible for SCA biosynthesis in *T. grandis*.**
**a** Overview of the fatty acid biosynthetic pathway. PDH pyruvate dehydrogenase, CT carboxyltransferase, BC biotin carboxylase, BCCP biotin carboxyl carrier protein, MCMT malonyl-CoA:ACP malonyltransferase, ACP acyl carrier protein, KAS ketoacyl-ACP synthase, SAD stearoyl-ACP desaturase, FATA acyl-ACP thioesterase A, FATB acyl-ACP thioesterase B, LACS long-chain acyl-CoA synthetase, DGAT diacylglycerol acyltransferase, PDAT phospholipid:diacylglycerol acyltransferase, PAP phosphatidic acid phosphatase, LPAT lysophosphatidic acid acyltransferase, GPAT glycerol-3-phosphate acyltransferase, CPT cholinephosphotransferase, FAD2 oleate desaturase, FAD3 linoleate desaturase, PC phosphatidyl choline. **b** *TgDES1* expression and SCA content in seeds from early development stage (May) to maturation stage (September). Different letters on the bars indicate statistical significance between samples at $\alpha = 0.05$ (one-way ANOVA and Tukey's test). Measurements were performed in three biological replicates and data are presented as mean + SD. **c** Expression of *TgELO1* and the content of its product cis-11,14-Eicosadienoic acid in seeds. Different letters on the bars indicate statistical significance between samples at $\alpha = 0.05$ (one-way ANOVA and Tukey's test). Measurements were performed in three biological replicates and data are presented as mean + SD. **d** Subcellular localization of TgDES1 and TgELO1 in *N. benthamiana* leaves. **e** Detection of SCA and its precursor in Arabidopsis Col-0 and the transgenic line overexpressing both *TgDES1* and *TgELO1*. Source data are provided as a Source Data file.

to be responsible for SCA biosynthesis in the basal eudicot *Anemone leveillei*[48]. Structure modeling of TgDES1, AtSLD2 and AL21 showed overall similar structures between TgDES1 and AtSLD2, particularly in the region where the active center was formed, whereas structure of AL21 was relatively diverged from TgDES1 (Fig. 4b). Since flowering plants rarely synthesize SCA, our phylogenetic and structural evidence suggested that this is possibly due to the loss of TgDES1 clade desaturases, while the ability of SCA biosynthesis in particular species of eudicots was largely attributed to the secondary gain of the $\Delta^5$-desaturase activity of evolutionarily independent counterparts. Similarly, close homologs of TgELO1 were not found in flowering plants but present in early land plants and algae, suggesting the co-evolution of $\Delta^5$-desaturase and $\Delta^9$-elongase in plants (Supplementary Fig. 13).

Characterization of protein sequences revealed the conservation of an N-terminal cytochrome b5-like domain and three histidine-rich boxes of TgDES1 clade desaturases (clade 1) and their two closely related groups (group 1 and group 2 of clade 2), whereas striking variation was observed in the first two histidine-rich boxes among different groups (Fig. 4c). A previous study reported that site-directed substitution of histidine boxes could influence substrate chain-length specificity and selectivity[50]. The single amino acid substitution likely directs the outcome of the desaturation reaction by modulating the distance between substrate fatty acyl carbon atoms and active center metal ions[51]. To test whether sequence variation of histidine-rich domains determined substrate specificity that led to the success of SCA biosynthesis, we replaced the histidine-rich domain of Arabidopsis desaturase AtSLD2 with that of TgDES1, and transiently expressed the construct in *N. benthamiana* leaves. We noted that *TgELO1* was not coexpressed with the engineered desaturase gene because 20:2$^{\Delta11,14}$-PC, the product of $\Delta^9$-elongase catalysis, could be detected in leaves of the wild-type tobacco. SCA was undetectable in *N. benthamiana* leaves expressing wild-type AtSLD2; however, switch of either of the two histidine-rich boxes from TgDES1 was sufficient to synthesize SCA in *N. benthamiana* leaves (Fig. 4d). Taken together, our data suggest that mutations in these two histidine-rich motifs of desaturases have led to the alternation of substrate specificity and consequently the evolution of specific clade for SCA biosynthesis, loss of which marks the significant metabolic diversity between gymnosperms and angiosperms.

## Dynamics of DNA methylation during seed development

Seed development in gymnosperms is a long process spanning multiple years[3]. To understand whether and how DNA methylation participates in seed development of *T. grandis*, as is evident in flowering plants[52], we profiled seed methylomes at three developmental stages (Fig. 5a; Supplementary Data 9). Genes involved in DNA methylation of all three cytosine contexts (CG, CHG, CHH) were identified in the *T. grandis* genome (Supplementary Data 3). The global average methylation levels of mCG, mCHG, mCHH in *T. grandis* seed genome were 83%, 69% and 4%, respectively. Both mCG and mCHG methylation levels were higher than those in most of previously studied angiosperms[53], coinciding with the proposal of positive correlation

between genome sizes and mCG/mCHG methylation levels[54]. mC of all sequence contexts was enriched at centromeric and peri-centromeric regions, despite that both mCG and mCHG were also broadly distributed in chromosome arms (Fig. 1b). In flowering plants, exons of genes are sometimes enriched with mCG but depleted with both mCHG and mCHH, which is referred to as gene body methylation (gbM)[55]. We observed the enrichment of mCG and depletion of mCHH in *T. grandis* genes; however, enrichment of mCHG was also found in transcribed regions (Fig. 5b and Supplementary Fig. 14a, b), which is similar to the pattern found in conifers[56]. GbM has been proposed to regulate gene transcription[55]. We observed a clear enrichment of mCG instead of mCHG/mCHH on moderately expressed genes, for which the expression was positively correlated with methylation levels (Fig. 5c and Supplementary Fig. 14c), indicating functional conservation of gbM in the sister lineage of angiosperms. Evolution of gbM is hypothesized to be associated with DNA methylation silencing of TEs in proximity of genes[55]. Consistently, we found that LTR-RTs, which were the major component of TEs in gene regions, were highly methylated (Fig. 5d), and that genes with TE insertions had both higher expression and CG methylation than those without TEs (Supplementary Fig. 15).

Seed development and germination genes are frequently localized within demethylation valleys (DMVs), where methylation level was low (e.g., <5%) for any of the cytosine contexts[57]. We identified 5099 common DMVs in the seed genome of the three samples, which spanned 30 Mb including the largest interval extending to 144 kb. The DMVs intersected with 4200 protein-coding genes, many of which encoded important classes of seed proteins, such as storage proteins, transcriptional factors, and enzymes for cell wall modification, hormone homeostasis and fatty acid biosynthesis (Fig. 5e, f). *T. grandis* seed is coated with a specialized outgrowth, called aril (Fig. 5a). During development, seed coat develops heavily lignified secondary cell walls to reinforce the outer surface of the seed[58], while preserves a soft inner one that directly surrounds the endosperm. Consistently, genes encoding laccases (n = 38), which function in cell wall lignification[59], and expansins (n = 13) that are associated with cell wall loosening[60], were frequently found in DMVs (Fig. 5f). Notably, 18% of *T. grandis* transcription factor (TF) genes (n = 370) were located within seed DMV regions, representing a significant enrichment ($\chi^2$ test; $P < 0.0001$; Supplementary Data 10). These TFs belonged to diverse gene families but were particularly abundant in MYB, NAC and AP2 families, which are known to regulate plant growth and development. The methylation of mCHH varied more remarkably than that of mCG and mCHG during seed development (Fig. 5g and Supplementary Fig. 16). We identified differentially methylated regions (DMRs) for each of the three cytosine contexts (Supplementary Data 11–13). Among genes overlapping with DMRs, 12% of them were differentially expressed, suggesting the translation of epigenetic variation to gene expression flexibility during seed development (Supplementary Fig. 17). GO enrichment analysis showed that DMR-associated genes were mainly enriched with those involved in photosynthesis and secondary metabolism (Supplementary Data 14), in line with the fact that photosynthesis fuels

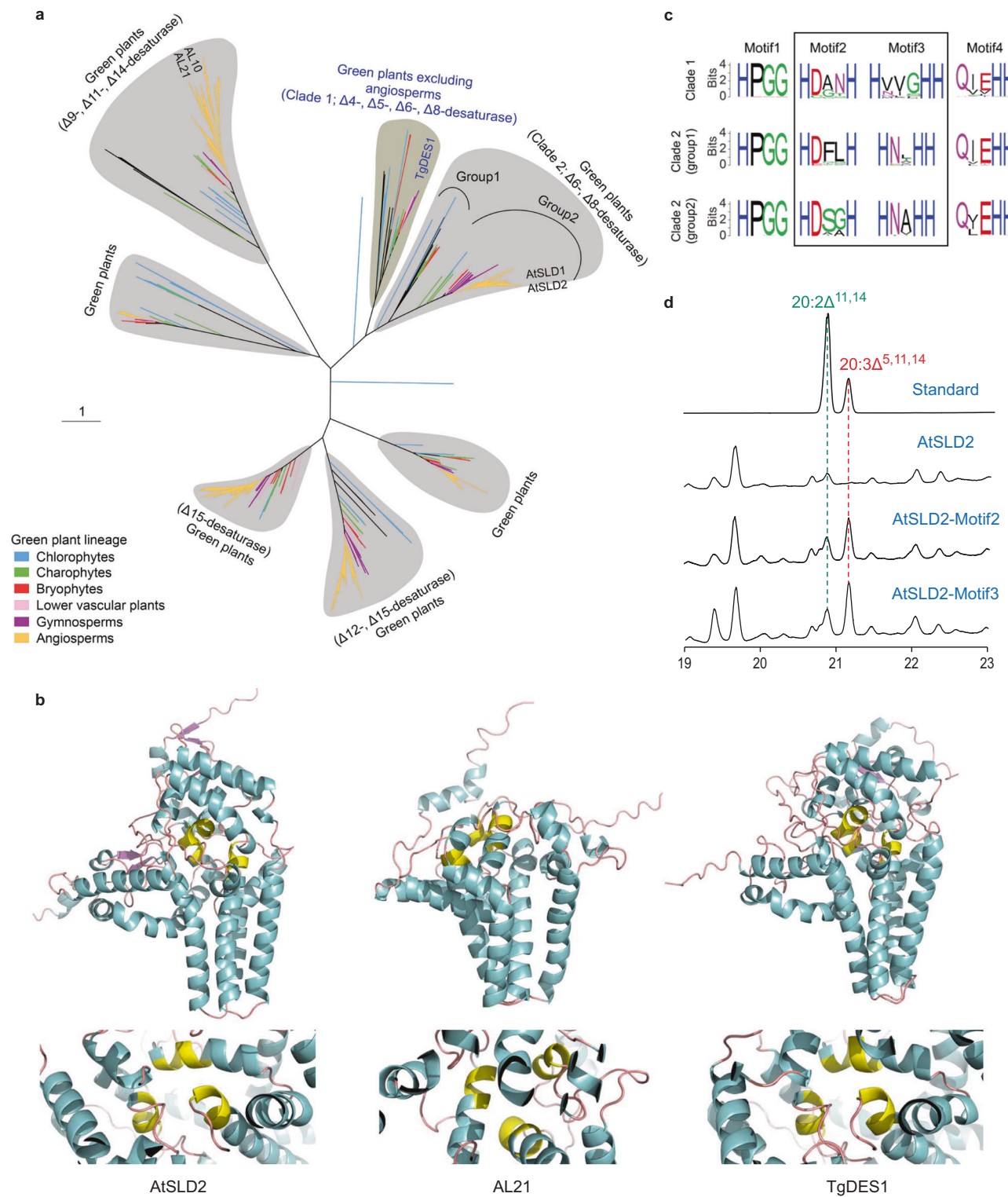

**Fig. 4 | Origin and evolution of plant Δ⁵-desaturases. a** Maximum likelihood phylogeny of plant desaturases. TgDES1 is clustered within a group (clade 1) close to a sister clade (clade 2) comprising Δ⁶- and Δ⁸- desaturases. **b** Structure modeling of TgDES1 and the desaturases from Arabidopsis (AtSLD2) and *Anemone leveillei* (AL21). Protein structures were modeled with AlphaFold2 and the bioactive center of each protein comprising three histidine-rich motifs is marked with yellow. **c** Comparison of conserved motifs in different desaturase groups showing in **a**. **d** Detection of SCA and its precursor in *N. benthamiana* leaves expressing Arabidopsis *AtSLD2* with conserved histidine-rich motifs (motif2 and motif3) replaced by those from *TgDES1* of *T. grandis*. AtSLD2, AtSLD2-Motif2, and AtSLD2-Motif3 are lines harboring Arabidopsis wild-type *AtSLD2* gene, *AtSLD2* with motif2 from *TgDES1*, and *AtSLD2* with motif3 from *TgDES1*, respectively.

energy-generating biochemical pathways by contributing oxygen to seed tissues during the development of green seeds as developing seeds suffer from limited penetration of oxygen, particularly into inner tissues[61].

## Discussion

Gymnosperms are considered as a treasure trove of life history on the earth. Here, we assembled a chromosome-level reference genome for the gymnosperm species *T. grandis*. The genome size is huge and

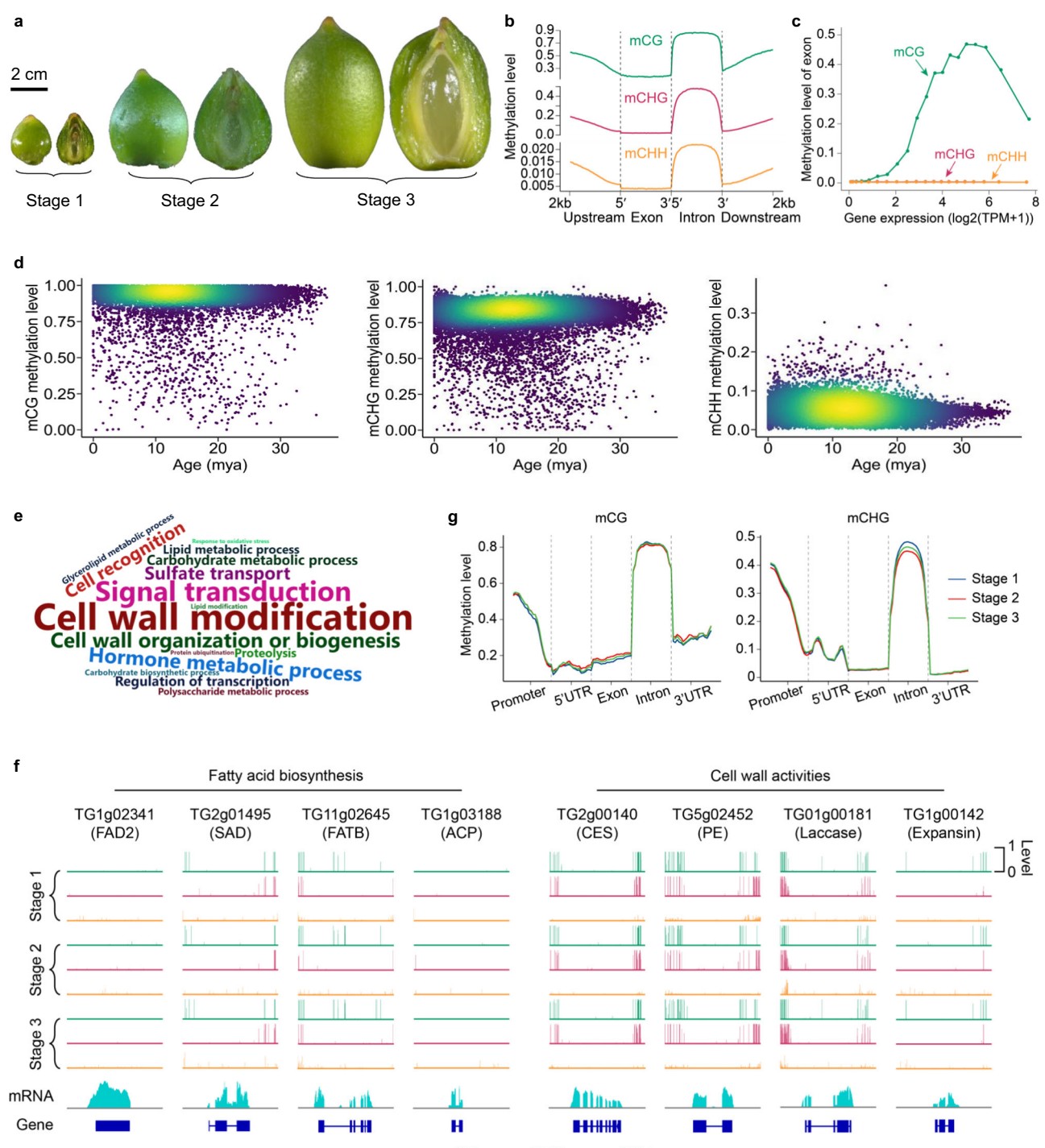

**Fig. 5 | *T. grandis* seed methylomes. a** Sampled seeds for methylome profiling. Pictures show the outer side and inner side (through longitudinal section) of the seeds. **b** Methylation levels of exon, intron and gene flanking regions in seeds. **c** Methylation levels at three cytosine contexts on exonic region of genes. Genes are categorized into 20 groups based on ordered expression levels. For each group, the median value of gene expression and the average methylation level across all exonic regions of genes are recorded. **d** Methylation levels of intact LTR-RTs in the *T. grandis* genome. **e** GO terms enriched in genes overlapping with demethylation valleys shared by seeds of all three developmental stages. GO terms with adjusted *P* value < 0.05 (two-sided Fisher's exact test with Benjamini–Hochberg correction) are plotted and sizes of GO terms in the word cloud figure correlate with their statistical significance. **f** View of methylation levels of selected genes overlapping with demethylation valleys. CES cellulose synthase, PE pectinesterase. **g** Comparison of mCG and mCHG methylation levels in different genomic regions of seeds at three stages. Source data are provided as a Source Data file.

much larger than most plant species ever sequenced. Based on this assembly and analysis of multi-omics data, we conclude that (1) accumulation of ancient LTR-RTs contributes to the bloating of the *T. grandis* genome, whereas *T. grandis* counteracts TE expansion through unequal recombination and epigenetic silencing with a mechanism potentially different from angiosperms; (2) gain or loss of important gene families in *T. grandis*, e.g., those involved in cell wall activities and paclitaxel biosynthesis, underly its phenotypic diversity, and MADS-box genes associated with reproductive organ identity include not only classical B- and C-function genes that are already proposed in

previous studies[35–39] but also additional genes (e.g., *TG8g01565*) that show expression pattern different from B- and C-function genes; (3) the Δ⁹-elongase and Δ⁵-desaturase are capable of synthesizing SCA, and these two enzymes have co-evolved and have been lost in flowering plants; moreover, substrate specificity of Δ⁵-desaturase is determined by the two histidine-rich boxes, mutation on which may lead to the alternation of substrate recognition, and subsequently the change of its product; (4) the seed genome of *T. grandis* comprises both heavily methylated repeat sequences and demethylation valleys, latter of which intersect with genes exerting important seed functions such as cell wall modification and fatty acid biosynthesis, as well as regulation of gene expression and hormone homeostasis. Overall, our high-quality reference genome coupled with comparative and functional genomic analyses provide insights into the gymnosperm biology, particularly in the biosynthesis and evolution of SCA that features metabolic versatility between the major land plant lineages.

## Methods

### Plant materials and sequencing

Young leaves from a plant of *T. grandis* grown in Shaoxing, China, were collected in March 2018 and used for DNA extraction following the CTAB (2%) method[62]. A paired-end (PE) library with insertion size of 350 bp was constructed using the Illumina Genomic DNA Sample Preparation kit following the manufacturer's instructions (Illumina), and sequenced on an Illumina NovaSeq system with a read length of 150 bp. A PacBio SMRTbell library was constructed using SMRTbell Express Template Prep Kit 2.0 and sequenced on a PacBio Sequel II platform. The circular consensus reads (HiFi reads) were generated using ccs software (https://github.com/pacificbiosciences/unanimity/) with parameter '-minPasses 3'. Hi-C library preparation and sequencing were performed by Novogene (Tianjin, China) following a protocol described elsewhere[63]. Briefly, libraries were prepared using leaf tissues fixed in 2% formaldehyde. Nuclei were extracted and permeabilized, and the chromatin was digested with DpnII restriction enzyme (NEB). The digested chromatin was blunt-ended and labeled with biotin. DNA ligation was performed using T4 DNA ligase (NEB), after which proteinase K was added for reverse crosslinking. The DNA fragments were then purified and sequenced on an Illumina NovaSeq platform with a read length of 2 × 150 bp.

To assist gene prediction, transcriptome sequencing was performed for samples collected from leaf, root, stem, young seed, aril, seed coat and kernel tissues of the same plant (Supplementary Data 1). Total RNA was extracted using TRIzol Reagent (Invitrogen) and quantified with NanoDrop ND-2000 spectrophotometer (NanoDrop Technologies). mRNA purified from total RNA with a RIN score ≥8 (Bioanalyzer 2100, Agilent Technologies) was used for library construction with the NEBNext Ultra II RNA Library Prep Kit for Illumina (NEB) following the manufacturer's instructions. The non-stranded RNA-Seq libraries were sequenced on an Illumina NovaSeq platform under 2 × 150-bp mode. For PacBio Iso-seq, total RNA from leaf, root, stem, aril and kernel tissues were pooled equally and cDNA was synthesized using the SMARTer PCR cDNA Synthesis Kit (Clontech). Size fractionation and selection (1–2, 2–3 and 3–6 kb) were performed using the BluePippin Size Selection System (Sage Science). The SMRT libraries were generated using the SMRTbell Template Prep Kit 1.0 (Pacific Biosciences) and sequenced on the PacBio RSII platform.

### Genome assembly and quality assessment

The HiFi reads were assembled using hifiasm[64] (version 0.8-dirty-r280) with default parameters and the assembled contigs were further polished by Racon (https://github.com/lbcb-sci/racon; v1.4.13) with Illumina reads. Purge Haplotigs[65] (version v1.1.0) was used to filter out redundant sequences in the assembly with parameters '-l 15 -m 70 -h 125' for the 'contigcov' subcommand and '-a 55' for the 'purge' subcommand. Illumina reads from Hi-C libraries were processed with Trimmomatic[66]

(v0.36) to remove adaptors and low-quality sequences. The cleaned reads were analyzed by HiCUP (https://www.bioinformatics.babraham.ac.uk/projects/hicup/) to identify non-duplicated valid alignments, which were then used for scaffolding with ALLHiC[67] (version 0.9.8). The initial scaffolding was manually curated using Juicebox (https://github.com/aidenlab/Juicebox). Completeness of the assembly was evaluated using the Illumina sequencing reads, which were mapped to the genome assembly using BWA-MEM[68].

### Repeat annotation and LTR insertion time estimation

Repetitive sequences were identified using a combination of homology-based and de novo predictions. A species-specific TE library for *T. grandis* was constructed to include LTR retrotransposons (LTR-RTs) and other TE elements identified by LTR_Finder[69] and RepeatModeler[70], respectively. This library was then combined with the Repbase library[71] for TE identification by RepeatMasker[72] (v.4.0.7). Repetitive elements were also predicted by RepeatProteinMask and the tandem repetitive sequences were identified by the TRF program[73]. To estimate LTR-RT insertion times, intact LTR-RTs were searched by LTR_Finder and LTRharvest[74]. MUSCLE[75] was used to align LTR sequences of intact LTR-RTs, and the nucleotide distance (*K*) between them was calculated with the Kimura two-parameter criterion using the distmat program in the EMBOSS package (http://emboss.sourceforge.net). The insertion time (*T*) was calculated as

$$T = K/2r \tag{1}$$

where the rate of nucleotide substitution (*r*) used for gymnosperm species was $2.2 \times 10^{-9}$ per base per year[11]. The putative centromeric repeats were determined based on the copy number and chromosomal distribution of the tandem repeats identified by TRF.

### Gene prediction and gene set assessment

Protein-coding genes were predicted using repeat-masked genome sequences. For homology-based prediction, protein sequences from one moss (*Physcomitrella patens*), one fern (*Selaginella moellendorffii*), seven angiosperms (*Amborella trichopoda, Arabidopsis thaliana, Oryza sativa, Phalaenopsis equestris, Populus trichocarpa, Vitis vinifera* and *Zea mays*), and four gymnosperms (*Ginkgo biloba, Gnetum montanum, Picea abies*, and *Pinus taeda*) were aligned to the *T. grandis* genome using TBLASTN[76] with an e-value cutoff of 1E−5. GenBlastA[77] was then applied to cluster adjacent high-scoring pairs from the same protein alignments, and the corresponding gene structures were identified with GeneWise[78] (v.2.4.1). Raw RNA-Seq reads were cleaned with Trimmomatic[66] (v0.36) and mapped to the *T. grandis* genome using TopHat2[79]. Subsequently, Cufflinks[80] (v.2.2.1) was employed to predict gene models. The cleaned RNA-Seq reads were also used to predict gene structures with Trinity[81] (v2.0.13) and PASA[82] (v2.2.0). All complete gene structures predicted by the PASA pipeline were used for gene model training for AUGUSTUS[83], GlimmerHMM[84], and SNAP[85]. These three predictors as well as geneid[86] and GENSCAN[87] were used for ab initio gene prediction with default parameters except that '-noInFrameStop=true -genemodel=complete' was applied to AUGUSTUS. Finally, all gene models predicted with different approaches were integrated to generate a high-confidence gene set using EVidenceModeler[88] with the following weight score matrix: PASA, 100; GeneWise, 20; Cufflinks, 20; AUGUSTUS, 5; other ab initio predictors, 1.

To evaluate the accuracy of predicted genes, we examined the coverage of highly conserved genes using BUSCO[19]. We further performed functional annotation of the *T. grandis* predicted gene models by searching against the databases Kyoto Encyclopedia of Genes and Genomes (KEGG; https://www.genome.jp/kegg/)[89], SwissProt and TrEMBL (https://www.uniprot.org/) using BLASTP with an e-value cutoff of 1E-5, and the best alignment hits were used to assign homology-based gene functions. GO (http://geneontology.org/)

categories and InterPro (https://www.ebi.ac.uk/interpro/) entries were obtained via InterProScan[90].

## Gene family evolution

The longest transcript of each of the protein-coding genes from 18 representative species (*Taxus wallichiana*, *Amborella trichopoda*, *Arabidopsis thaliana*, *Ginkgo biloba*, *Gnetum montanum*, *Welwitschia mirabilis*, *Oryza sativa*, *Solanum lycopersicum*, *Physcomitrella patens*, *Pinus tabuliformis*, *Selaginella moellendorffii*, *Vitis vinifera*, *Sequoiadendron giganteum*, *Azolla filiculoides*, *Klebsormidium flaccidum*, *Chara braunii*, *Marchantia polymorpha* and *Penium margaritaceum*) and *T. grandis* were selected to construct gene families based on all-against-all BLASTP alignments using OrthoFinder[91]. Phylogenetic analyses were conducted using IQ-TREE[92] (v. 2.1.3). Based on MRCA analysis using CAFE[93] (v.4.2.1), we determined the expansion and contraction of gene families between extant species and their last common ancestors.

## Analysis of WGD events in the *T. grandis* genome

All-against-all BLASTP search was performed with an e-value cutoff of 1E-5. The top five alignments were selected for each gene and used to detect syntenic gene pairs located in collinear blocks with MCScanX[94]. Paralogous gene pairs were determined by the best reciprocal BLASTP alignments. $K_s$ of each syntenic or paralogous gene pair was calculated using YN00 in the package PAML 4.8a[95] with default parameters. Phylogeny based inference of WGD was carried out based on the reconciliation of each gene tree and the species tree.

## Small RNA sequencing

Total RNA (3 μg) from leaves was isolated for small RNA library construction using NEB Next® Multiplex Small RNA Library Prep Set for Illumina® (NEB, USA) following manufacturer's recommendations. DNA fragments in the constructed library within the range of 140-160 bp were recovered and the library was assessed on an Agilent Bioanalyzer 2100 system and subsequently sequenced on an Illumina HiSeq 2500 platform. Raw reads of small RNA library were processed with Trimmomatic[66] (v0.36) to remove adapters and then aligned to the reference genome using Bowtie[96] with no mismatch allowed.

## Whole-genome bisulfite sequencing

*T. grandis* seeds were collected from a single tree on March 8 (stage 1), March 24 (stage 2) and April 8 (stage 3) of the year 2021 for bisulfite and transcriptome sequencing. About 100 ng high-quality genomic DNA spiked with 0.5 ng lambda DNA were sonicated with Covaris S220 (parameters: PIP, 50 W; duty factor, 20; cycles per burst, 200; treatment time, 110 s; temperature, 20 °C; sample volume, 52 μL). The fragmented DNA (200–300 bp) were treated with bisulfite using EZ DNA Methylation-GoldTM Kit (Zymo Research), and the library was quality assessed and sequenced on the Illumina NovaSeq platform with the paired-end mode.

Raw reads were cleaned with Trimmomatic[66] (v0.36) to remove adaptors and low-quality sequences. To align the cleaned reads, both the reference genome and reads were transformed (C-to-T and G-to-A) and then aligned with Bismark[97] (version 0.16.3) with parameters "-X 700 −dovetail". Reads that produced a unique best alignment against both "Watson" and "Crick" strands of the genome were kept and the methylation state of all cytosine nucleotides were inferred. The sodium bisulfite conversion rate was estimated based on the read alignments to the lambda genome. Methylated sites were identified with a binomial test using the methylated counts (mC), total counts (mC+umC) and the conversion rate (r). Sites with FDR-corrected $P$ value < 0.05 were considered as methylated sites. To calculate whole-genome methylation level, we divided the genome into 10-kb bins, and the methylation level of each window was calculated as count(mC)/(count(mC) + count(umC)). Differentially methylated regions (DMRs) were identified using the DSS software[98] under the $P$ value threshold of 0.05. DMRs were cataloged based on whether and how they overlapped with genes. Continuous cytosine sites across the *T. grandis* genome with methylation level <5% in any contexts were merged and defined as demethylation valleys.

## Transcriptome sequencing and differential gene expression analysis

Male cones were collected from the *T. grandis* tree at eight different stages during February and April of 2021 with a time interval of 7 days, and female cones were collected at six different stages during January and April of 2021 with a time interval of 16 days. Other samples including kernel, leaf, root, and stem were collected from the same tree. Each sampling was performed with three biological replicates. Total RNA was extracted using TRIzol Reagent (Invitrogen). RNA-Seq libraries were constructed with the NEBNext Ultra II RNA Library Prep Kit for Illumina (NEB) following the manufacturer's instructions and sequenced on an Illumina NovaSeq platform under 2 × 150-bp mode. Raw RNA-Seq reads were cleaned using Trimmomatic[66] (v0.36). The cleaned reads were mapped to the genome using STAR aligner[99] (v2.7.10a). The alignments were counted using HTSeq-count[100] and differentially expressed genes were identified with DESeq2 (ref. [101]) under the cutoff of adjusted $P \le 0.01$ and fold change ≥2.

## HGT identification

Potential HGTs were identified based on homology scores and phylogeny signals[102]. Briefly, we created three customized databases, namely an out-group database comprising all protein sequences from archaea, bacteria and fungi, an in-group database including protein sequences from 10 published gymnosperm species, and a mid-group database consisting of sequences from all published plants excluding gymnosperms. Protein sequences of *T. grandis* were blasted against the three customized databases separately with an e-value cutoff of 1E-5. For each query protein sequence, we preserved no more than 100 blast hits (one hit per species) for each database and calculated the average bit-score value (ABV) of the alignments. Query proteins with the ABV of out-group larger than that of mid-group were retained. We performed rigorous phylogenetic analyses for each of remaining query proteins and manually inspected the topology of the tree. *T. grandis* genes supported by both ABV and phylogeny were considered as potential horizontally transferred genes.

## Fatty acid determination

About 0.5 g dried samples were mixed with 9 mL 10% $H_2SO_4$-$CH_3OH$ solution at room temperature for 10 h. The fatty acid methyl esters were filtered and then extracted with 30 mL distilled water and 30 mL dichloromethane. The organic phase was dried with anhydrous sodium sulfate and concentrated to about 1 ml with a nitrogen blower. The concentrated extract was used for fatty acid analysis by gas chromatography (GC; Thermo Scientific TRACE-1300, Italy) with the methyl fatty acid used as an internal standard. GC separation was performed using an Agilent DB-WAX capillary GC column (30 m × 0.25 mm and 0.25 μm film thickness), and 1 μl of each sample was injected in split mode with a 1:20 ratio. Ultrapure helium was used as the carrier gas. The injection port and detector temperatures were set at 220 °C and 240 °C, respectively. The column temperature programming started at 140 °C held for 1 min and heated up to 250 °C at a rate of 4 °C/min. The column temperature was held for 2 min at 250 °C.

## Subcellular localization

The CDS of each gene, without stop codon, was cloned and fused to the N-terminus of the GFP gene of the pCAMBIA1300-GFP vector. The

resultant plasmid was introduced into *Agrobacterium tumefaciens* GV3101. Positive clones were incubated to an $OD_{600}$ of 0.6, and then centrifuged at 8000 rpm for 6 min. Collected cells were resuspended with infiltration buffer (10 mM $MgCl_2$, 0.2 mM acetosyringone, and 10 mM MES at pH 5.6), which was then injected into the leaves of *Nicotiana benthamiana*. After 3-d of culture, GFP fluorescence signal from the leaves was observed and captured using confocal laser scanning microscopy (LSM510: Karl Zeiss).

## Quantitative real-time PCR analysis

Total RNA was extracted using the RNAprep pure Plant Kit (TIANGEN). First-strand cDNA was synthesized from 1 µg of total RNA using the PrimeScript™ RT Master Mix Kit (Takara). SYBR Premix Ex Taq™ kit (Takara) was used to perform quantitative real-time PCR. Expression data of target genes were corrected with the expression of actin encoding gene. The reaction conditions were 95 °C for 10 s, 55 °C for 10 s, 72 °C for 20 s, 45 cycles. The relative expression was calculated using $2^{-\Delta\Delta Ct}$ method.

## Heterologous synthesis of SCA in Arabidopsis seeds and *N. benthamiana* leaves

Coding regions of *TgEOL1*, *TgDES1*, *AtSLD2* (*AT2G46210*) and two recombinant genes (*AtSLD2-Motif2* and *AtSLD2-Motif3*) were inserted into the downstream of the 35S promoter of the binary vector (pCAMBIA1300), respectively. Each of the resulting constructs was transformed into *Agrobactrium tumefaciens* strain GV3101, which was then grown at 28 °C in LB medium supplemented with kanamycin (50 mg/L) and rifampicillin (50 mg/L) until $OD_{600}$ reaching 0.6. For transient expression of *AtSLD2* and two recombinant genes in *N. benthamiana* leaves, cells were harvested and resuspended in 10 mM MES buffer (containing 10 mM $MgCl_2$ and 0.1 mM acetosyringone) to a final OD600 of 1.0. The cells of each strain were infiltrated into the young leaves of five-week-old *N. benthamiana* plants using a needleless syringe, which were harvested 5 days later for measurement of SCA content. For generation of *TgDES1* and *TgELO1* overexpressed Arabidopsis, the pCAMBIA1300-*TgELO1* and pCAMBIA1300-*TgELO1* constructs were transformed into Arabidopsis via *A. tumefaciens*-mediated floral dip method. Hygromycin-resistant T1 plants were planted for seed harvesting, and T2 seeds with a hygromycin resistance ratio of 3:1 were selected to collect T3 seeds. T3 seeds with 100% resistance to hygromycin were used for determination of SCA content.

## Reporting summary

Further information on research design is available in the Nature Portfolio Reporting Summary linked to this article.

# Data availability

Genome assembly and raw reads for genome, transcriptome and methylome sequencing have been deposited in the National Center for Biotechnology Information BioProject database under accession PRJNA938254 and the CNGB Sequence Archive (CNSA) of China National GeneBank DataBase (CNGBdb) under accession CNP0003453. Genome assembly and annotation are also available at Figshare [https://doi.org/10.6084/m9.figshare.21089869]. Source data are provided with this paper.

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

## Acknowledgements

This research was supported by grants from the National Natural Science Foundation of China (NSFC) to J.W. (grant no. U20A2049), L.S. (grant no. 31971699), X.S. (grant no. 32102318), the Key Research and Development Program of Zhejiang Province to H.L. (grant no. 2021C02001), the Scientific Research Startup Fund Project of Zhejiang A&F University to H.L. (grant no. 2018FR028), and the State Key Laboratory of Subtropical Silviculture grant to J.W. (grant no. ZY20180312 and ZY20180209). The authors thank Dr. Emily E.D. Coffey from Atlanta Botanical Garden (USA) and Prof. Mark W. Schwartz from University of California, Davis for providing plant samples.

## Author contributions

J.W., X.S., H.L. and L.S. conceived and supervised the project. W.C., Y.G. and S.Z. collected samples and performed transgenic experiments. X.S., X.L. and H.Z. constructed libraries and performed bioinformatics analysis. X.S. and H.L wrote the manuscript. Z.F. and J.W. revised the manuscript.

## Competing interests

The authors declare no competing interests.
