## [Peer Review File · Nature Communications]

The *Torreya grandis* genome illuminates the origin and evolution of gymnosperm-specific sciadonic acid biosynthesisReviewers' Comments:

Reviewer #1:

Remarks to the Author:

Review of manuscript NCOMMS-22-43912-T

The manuscript by Lou et al. reports a de novo chromosome-scale genome assembly and annotation for the gymnosperm species *Torreya grandis* (Chinese nutmeg). This work is a significant addition to the other two recently published genome assemblies in the Taxaceae family (*Taxus wallichiana* and *Taxus chinensis*). Given the complexity, abundance of repetitive elements and large genome sizes of gymnosperm genomes, it is noteworthy that the authors were able to produce a chromosome-scale assembly as only 7 gymnosperm genome assemblies with that level of contiguity have been published so far (genome sizes varied from 8 to 25 Gb).

Repetitive content (59.8%) in *T. grandis* is reported to be lower than in other gymnosperms (up to ~85%) but introns (mainly composed by LTRs) were longer. The authors suggest that the combination of a high ratio of solo:intact LTRs, presence of homologs of key components of the RdDM pathway but low frequency of 24-nt sRNAs implies the presence of alternative pathways for transposon silencing in *T. grandis* and other gymnosperms compared to flowering plants. A higher ratio of solo:intact LTRs was previously reported in *Pinus tabulaeformis* (Niu et al 2022). Alternative pathways for transposon silencing have been postulated earlier by other authors to explain the large genome sizes and the extensive number of repetitive elements in gymnosperm species, however no conclusive evidence has been published so far. Additional evidence is required to conclude that the patterns observed in *T. grandis* can be generalized to other gymnosperms (especially considering its lower %repetitive content). Also, the evidence for the observation that 24-nt are less abundant than 21-nt comes from the analysis of needle tissue (suppl figure 3) when it has been reported that 24-nt are more abundant in developing seedlings and basal meristems in other gymnosperm species (such as *Welwitschia* and others). More evidence from other tissues is required to confirm the (potentially different) small RNAs patterns in gymnosperms.

Some important results of this work include the analysis of whole genome DNA methylation at different seed developmental stages. Highly repetitive gymnosperms genomes are also highly methylated, however very few studies have looked at whole genome DNA methylation in gymnosperms mostly due to the technical and computational complexities of working with huge genomic datasets. Global methylation levels found in this study are consistent with previous studies in gymnosperms and are significantly larger than estimates in flowering plants. Higher methylation levels are found in promoter and intronic regions, which supports recent evidence of the importance of methylation in gene regulation in gymnosperms. Other interesting results are the overrepresentation of photosynthetic and secondary metabolism gene ontologies in genes of differentially methylated regions, and the role of methylation in regulating cell wall modification and fatty acid biosynthesis in seeds.

The authors have performed significant work to identify genes of the sciadonic acid biosynthesis, which might provide the baseline for trait improvement in this economically important gymnosperm species in China.

The manuscript is well-written, although the Introduction could be better developed to include recent work in other gymnosperms and to define the research objectives. The methodology is sound, well-detailed, and exceeds the expected standards for work in the field. No major flaws have been observed.

Reviewer #2:

Remarks to the Author:

This study used abundant HiFi long reads for assembly of the genome of *T. grandis* and the overall quality is OK since the genome size is large. I have a few comments for further improvement of the manuscript.

1. In introduction, there is lack of description of progress of biosynthesis of SCA, especially the pathway and if the key genes have been cloned and functionally characterized (desaturase and elongase). Have C18 Δ 9-elongase and C20 Δ 5-desaturase genes been identified in *T. grandis*? If not, authors should indicate this in introduction. Based on the authors' description between line 63 and 66, they firstly discovered these two enzymes.
2. How about the mapping rate of the RNA-seq? Author should present this which is an indication of the quality of the genome.
3. Line 261 to 270, It is not necessary to analyze this since Paclitaxel and relevant metabolites were not detected in *T. grandis*. Besides, Supplementary Figs. 11 to 13 are not clear and do not provide key information. It is better to focus on SCA story.
4. The results of Dynamics of DNA methylation during seed development is somehow separate from previous results. Authors may want to provide more data and strength for this work. Most importantly, I do not see the reason to analyze cell wall-related activity here. I suggest authors may focus on major activities/pathways in the seed, such as fatty acid/oil pathway.

Minor:

Line 573, the method is too simple. There is not cited reference and no detail for fatty acid analysis, such as GC parameters.

Line 48, In all kinds of fatty acids,

Line 49, the content of SCA is over 10% in the kernel oil. When we talk about oil, we usually use oil content. When we talk about fatty acid, we usually use fatty acid composition (ratio of a specific fatty acid)

Line 78, Two and nine chromosomes were, do you mean number two and number 9 chromosomes? In addition, we cannot see telomeric sequences (5'-TTTAGGG-3') at both and single ends of the chromosomes. It is not necessary to cite Fig. 1b here.

Line 289, known desaturases and elongases? Elongase should be mentioned here.

Reviewer #3:

Remarks to the Author:

Lou and colleagues successfully assembled *T. grandis* with such large genomes and elucidated seed development and gymnosperm-specific sciadonic acid biosynthesis through bioinformatics analysis. It is also very important to pay attention to the DNA methylation of large genomes. I was very interested to read this manuscript, but I still have some major concerns need to be clarified.

Major concerns:

1. In lines 132 to 136, I was confused about the analysis in this part, especially the collinearity analysis couldn't tell me that *T. grandis* experienced two WGD events after the differentiation from *G. montanum*. Because the collinearity analysis failed to show 1:2(*G. montanum*: *G. biloba*) and 1:4 (*G. montanum*: *T.G. randis*) (Figure 1e). in addition, details of supplementary5 are also vague, hoping to elaborate on their views in more details.
2. What percentage of genes have TE insertions, and whether these insertions occurred at intron or exon, or exon and intron coexist, their methylation levels, and whether there is an effect on host gene expression, and what effect?
3. Figure 5d shows the methylation and insertion of intact LTRs. How does this relate to line 390-391.
4. For DNA methylation analysis, I did not see the summary of sequencing depth and alignment ratio, conversion rate etc. I think the quality of methylation data is very important for the subsequent

analysis. We know that it is very challenging to perform DNA methylation analysis of such a large genome by using second-generation sequencing of WGBS.

5. The authors failed to find abundant small 24nt RNAs in leaves, which may be because the sampling was not complete enough to infer functional differences of the RdDM pathway in gymnosperms.

Minors:

1. The methods section, please add more methylation analysis methods and software parameters, including DMV, DMR recognition and so on

Reviewer #4:

Remarks to the Author:

This work details a high-quality chromosome-scale genome assembly that is an important and useful addition to the available genomics resources for conifers. The methods applied appear to be appropriate but there are additional methods details and summary analyses that I would like to see to establish further confidence in the assembly accuracy, in particular scaffolding results and haplotype purging, and annotation completeness. I appreciate that those details not being included is likely due to word limitations for the article type, but I do think a more extensive and complete methods description in the form of a supplementary note would be useful. One aspect that I find surprising is that the ab initio gene predictions tools are all described as working out of the box. This has not at all been my own experience with conifer genome, so I am curious to know if there are some details and filtering steps that were required that are not reflected in the current methods description. It would also have been interesting to have some comparison to, and comment on, the differences to the results recently reported for the Chinese pine genome. There is a large difference in the reported number of annotated protein coding genes as well as noticeable differences in BUSCO scores. As these differences can influence the results of analyses of gene families and evidence for WGD events. Similarly, incomplete haplotig purging and influence such analyses. I was also surprised by the difference reported here for the solo to complete LTR ratio as this is in contrast to previous reports in conifers. Some brief comment on this in the conclusions would be interesting to include.

L34 The sentence should be in the past tense – so marked not marks.

L36 'diverged in the'

L37 earth should be Earth

L38 I do not understand what is meant by this sentence as it does not link to anything previously written. Apparent based on what?

L42 They are not nuts, they are seeds.

L57 It is largely due to their large genomes sizes and high repeat content, not heterozygosity (there are numerous sequenced heterozygous angiosperms, for example).

L60 Anchored how?

L71 About 96X and 23.2X (the decimal is not needed when using the qualifier 'about').

L76 A citation is needed for the 101 bp centromeric repeat.

L82 It would be good to state the number of these that are complete and that are not duplicated.

L108 Comparing to other gymnosperms with low contiguity assemblies is not appropriate. Intron sizes

in all conifers appear to be similar (e.g. in Chinese pine, which is probably the other highest quality conifer assembly currently available).

L112 This observation is in contrast to the results that have been reported for Norway spruce. Do the authors have an explanation for these contrasting results? A stronger force for removal runs counter to the large genome size of *T. grandis*.

L117 It should be noted here that previous observation in Norway spruce identified that 24 nt sRNA expression is more dynamic and tissue dependent with results dependent on that sample assayed.

L124 The spelling of paralog and orthologs is inconsistent here compared to the legend of Figure 1.

L132 Are the authors confident that all genome assemblies and annotations used for this analysis are of sufficient (and comparable) quality to have confidence in these results? A number of factors, including the efficiency of haplotype purging, could influence the result.

L149 This sentence needs to point to where the result is reported, e.g. a table or figure (or a citation).

L159 As gymnosperms do not have true flowers this subheading is maybe not an ideal choice of wording

L161 Homolog spelling

L256 VND, NST and SND are discussed in the NorWood wood development publication profiling gene expression during wood development in Norway spruce. These results would perhaps benefit from comparison to this previous report.

L261 As *T. grandis* does not produce Paclitaxel I see little value in this section – although there is nothing wrong with the results reported here.

L294 This result does not show that the genes are required in *T. grandis* as this would require a knock-out. It shows that they are capable of this.

Was there correspondence between the DMVs/DMRs and 21 nt sRNAs? What was the methylation status of repeats in the genome and were those consistent throughout seed development?

Conclusion 3 is not extensively supported by analyses presented in this paper. Similarly, while conclusion 2 is inline with generalizations, this is also not examined in detail with only gross-scale patterns reported here. For example, there is no examination of the relationship between methylation status and whether LTRs are present as complete or solo elements.

For conclusion 6 to be more fully evaluated would also require methylation data from non seed samples. It could be that these low methylation regions and more generally low methylation comply because they contain genes. There is no statistical support presented for these results.

L437 I do not follow how this genome specifically fills a gap when other genomes for gymnosperms and conifers are already available. There are some compelling results in this paper specific to *T. grandis* but few new or novel results more generally.

The conclusion would benefit from comparison to results from existing conifer genome publications, including work examine methylation patterns, differential expression and sRNA expression. This work is not performed in a void of available information, yet the conclusion section does not reflect this.

L444 The DNeasy kit does not extract truly HMW DNA and would not normally be recommended for

use with PacBio long reads (although HiFi reads are admittedly not super long).

L460 Was RNA integrity assessed with e.g. a Bioanalyzer?

L461 How was rRNA depleted and what primers were used for cDNA synthesis?

L463 Were the data strand specific or not?

L473 What parameters were set for Purge Haplotigs and how was the success of purging determined?

L474 Were PCR duplicates removed (if so, how?) and was library complexity assessed?

L477 Assessed how?

L481 What knowledge does this refer to?

L496 These tools all need parameter settings to be defined. Some also require a gold standard set of genes to be used for model training. How was this performed? These tools are known to perform poorly in conifer genomes due to the presence of long introns so how was this overcome? It is also very common that these ab initio methods will identify some additional repeats and pseudogenes, so was, or how was, this assessed? Were any filters applied for e.g. minimum gene or CDS length?

L507 Was BUSCO applied to the set of predicted proteins or to the genome, or both?

L543 Describe the sample used for this DNA extraction.

How many genes were derived from the different evidence types and how many were commonly supported? The information presented about this is very minimal.

The methods do not detail expression analysis of the RNASeq data.

Figure 1. Why are different track types used for the Gypsy and Copia elements?

Figure S1 The figure is quite low resolution so it is hard to view in detail but there are some features in the contact map that appear to be potential miss-assemblies. How was this assessed and was any manual curation performed after the initial Hi-C scaffolding?

Figure S3 While 24nt sRNA are relatively low abundance it would be interesting to know which genomic features these aligned to. Similarly, is there a population of 21nt sRNAs that are repeat associated and that correspond to methylation regions?

Figure S17 This is not a correlation

Reviewer #1 (Remarks to the Author):

The manuscript by Lou et al. reports a *de novo* chromosome-scale genome assembly and annotation for the gymnosperm species *Torreya grandis* (Chinese nutmeg). This work is a significant addition to the other two recently published genome assemblies in the Taxaceae family (*Taxus wallichiana* and *Taxus chinensis*). Given the complexity, abundance of repetitive elements and large genome sizes of gymnosperm genomes, it is noteworthy that the authors were able to produce a chromosome-scale assembly as only 7 gymnosperm genome assemblies with that level of contiguity have been published so far (genome sizes varied from 8 to 25 Gb).

Response: We thank the reviewer for the positive comments.

Repetitive content (59.8%) in *T. grandis* is reported to be lower than in other gymnosperms (up to ~85%) but introns (mainly composed by LTRs) were longer. The authors suggest that the combination of a high ratio of solo:intact LTRs, presence of homologs of key components of the RdDM pathway but low frequency of 24-nt sRNAs implies the presence of alternative pathways for transposon silencing in *T. grandis* and other gymnosperms compared to flowering plants. A higher ratio of solo:intact LTRs was previously reported in *Pinus tabulaeformis* (Niu et al 2022). Alternative pathways for transposon silencing have been postulated earlier by other authors to explain the large genome sizes and the extensive number of repetitive elements in gymnosperm species, however no conclusive evidence has been published so far. Additional evidence is required to conclude that the patterns observed in *T. grandis* can be generalized to other gymnosperms (especially considering its lower %repetitive content).

Response: Thanks for the comments. Genomes of gymnosperms are generally large, which is mainly contributed by the expansion of LTR-RTs. In contrast to many small-genome angiosperms, of which LTR-RTs are more recent (<4 mya), gymnosperms accumulate relatively ancient LTR-RTs (10-30 mya). We speculate that continuous elimination without remarkable expansion of LTR-RTs may have contributed to the high solo:intact LTR ratios in gymnosperms, including *T. grandis* (solo to intact LTR ratio: 4.3), *Taxus wallichiana* (5.5), *Ginkgo biloba* (4.26), *Welwitschia mirabilis* (3.87), and *Gnetum montanum* (2.07). Both our data (please also check our response to the next question) and previous studies provide evidence for the divergent patterns of TE elimination and TE-derived sRNA processing between angiosperms and gymnosperms. We believe the mechanisms underlying such differences are interesting and deserve a deep exploration in the future. Nonetheless, we have rephrased the corresponding sentences in the revised manuscript (Lines 101-106 and 110-113).

Also, the evidence for the observation that 24-nt are less abundant than 21-nt comes from the analysis of needle tissue (suppl figure 3) when it has been reported that 24-nt are more abundant in developing seedlings and basal meristems in other gymnosperm species (such as *Welwitschia* and others). More evidence from other tissues is required to confirm the (potentially different) small RNAs patterns in gymnosperms.

Response: Thanks. According to the reviewer's suggestion, we have generated sRNA sequences for seven additional samples, and the tissues used for sRNA sequencing now include leaf, female cone, male cone (two stages), kernel, stem, root tip, and aril. As shown in Supplementary Fig. 3, 21-nt sRNAs were indeed the most abundant in all these tissues, while productions of 22-nt and 24-nt sRNAs were tissue-specific. This pattern is similar to that observed in conifers (Nakamura et al., BMC Genomics, 2019, 20:997; Niu et al., Cell, 2022, 185:204-217) and *Welwitschia*

mirabilis (Wan et al., Nat Commun., 2021, 12:4247), and implies the potential difference of sRNA processing between gymnosperms and angiosperms.

Some important results of this work include the analysis of whole genome DNA methylation at different seed developmental stages. Highly repetitive gymnosperms genomes are also highly methylated, however very few studies have looked at whole genome DNA methylation in gymnosperms mostly due to the technical and computational complexities of working with huge genomic datasets. Global methylation levels found in this study are consistent with previous studies in gymnosperms and are significantly larger than estimates in flowering plants. Higher methylation levels are found in promoter and intronic regions, which supports recent evidence of the importance of methylation in gene regulation in gymnosperms. Other interesting results are the overrepresentation of photosynthetic and secondary metabolism gene ontologies in genes of differentially methylated regions, and the role of methylation in regulating cell wall modification and fatty acid biosynthesis in seeds.

Response: Thanks for the nice recapitulation of the significance of the study!

The authors have performed significant work to identify genes of the sciadonic acid biosynthesis, which might provide the baseline for trait improvement in this economically important gymnosperm species in China.

Response: Thanks for the nice recapitulation of the significance of the study!

The manuscript is well-written, although the Introduction could be better developed to include recent work in other gymnosperms and to define the research objectives. The methodology is sound, well-detailed, and exceeds the expected standards for work in the field. No major flaws have been observed.

Response: Thanks! The introduction has been revised according to the reviewer's suggestion (Lines 55-64).

Reviewer #2 (Remarks to the Author):

This study used abundant HiFi long reads for assembly of the genome of *T. grandis* and the overall quality is OK since the genome size is large. I have a few comments for further improvement of the manuscript.

Response: We thank the reviewer for the positive comments.

1. In introduction, there is lack of description of progress of biosynthesis of SCA, especially the pathway and if the key genes have been cloned and functionally characterized (desaturase and elongase). Have C18 Δ 9-elongase and C20 Δ 5-desaturase genes been identified in *T. grandis*? If not, authors should indicate this in introduction. Based on the authors' description between line 63 and 66, they firstly discovered these two enzymes.

Response: The Δ 9-elongase and Δ 5-desaturase have not been functionally described in gymnosperms prior to this study. We previously searched the transcriptome dataset of *T. grandis* to identify transcripts associated with SCA biosynthesis (Wu et al., Ind Crops Prod., 2018, 120:47-60); however, the present study was the first to confirm these genes and their functions of Δ 9-

elongase and $\Delta 5$ -desaturase, respectively, in the SCA biosynthesis. We have clarified this in the introduction of the revised manuscript (Line 63-64).

2. How about the mapping rate of the RNA-seq? Author should present this which is an indication of the quality of the genome.

Response: Thanks for the suggestion. The average mapping rate of RNA-Seq data of 67 different libraries is 95.5% (median: 96.2%; highest: 97.5%), which confirms the high quality of the genome assembly. The detailed mapping rates of the RNA-Seq data have been provided in Supplementary Table 1 and we also mentioned the mapping rate in the manuscript (Line 85).

3. Line 261 to 270, It is not necessary to analyze this since Paclitaxel and relevant metabolites were not detected in *T. grandis*. Besides, Supplementary Figs. 11 to 13 are not clear and do not provide key information. It is better to focus on SCA story.

Response: Thanks for the suggestion. We have removed this short paragraph on paclitaxel biosynthesis as well as Supplementary Figures 11-13 in the revised manuscript. However, since *Torreya* is the second genus in Taxaceae with a genome assembly, readers might be interested in whether the *T. grandis* genome encodes paclitaxel biosynthetic genes; therefore, we have added one sentence to briefly address this issue in the subsection of “Gene family evolution” (Lines 152-154).

4. The results of Dynamics of DNA methylation during seed development is somehow separate from previous results. Authors may want to provide more data and strength for this work. Most importantly, I do not see the reason to analyze cell wall-related activity here. I suggest authors may focus on major activities/pathways in the seed, such as fatty acid/oil pathway.

Response: Thanks for the comments. The pattern of DNA methylation for genome size evolution and organ development has been well documented in flowering plants; however, whether the rules in angiosperms can be generalized to non-flowering plants are far from being conclusive, particularly in the context that genomes of gymnosperms are much more complex and are also presently underrepresented. Therefore, our results on the dynamics of DNA methylation make important contributions to the understanding of the pattern and potential function of DNA methylation in non-flowering plants. The significance of DNA methylation analysis has been acknowledged by the other two reviewers, which we quote here: Reviewer 1 “Some important results of this work include the analysis of whole genome DNA methylation at different seed developmental stages. Highly repetitive gymnosperms genomes are also highly methylated, however very few studies have looked at whole genome DNA methylation in gymnosperms mostly due to the technical and computational complexities of working with huge genomic datasets.”, and Reviewer 3 “It is also very important to pay attention to the DNA methylation of large genomes”.

The development and maturation of *T. grandis* seeds are accompanied by morphological changes and nutritional conversion. Through DNA methylation analysis, we found that genes associated with cell wall biogenesis (which controls seed morphology as cell expansion requires activities of cell wall related enzymes) and fatty acid biosynthesis are epigenetically regulated. These findings are new and relevant, which we believe will contribute to a better understanding of the biology of *T. grandis* and other gymnosperms.

Minor:

Line 573, the method is too simple. There is not cited reference and no detail for fatty acid analysis, such as GC parameters.

Response: The detailed method for fatty acid analysis has been added in the revised manuscript (Lines 535-542).

Line 48, In all kinds of fatty acids,

Response: This sentence has been revised (Line 47-49).

Line 49, the content of SCA is over 10% in the kernel oil. When we talk about oil, we usually use oil content. When we talk about fatty acid, we usually use fatty acid composition (ratio of a specific fatty acid)

Response: Thanks. This sentence has been revised (Lines 47-49).

Line 78, Two and nine chromosomes were, do you mean number two and number 9 chromosomes? In addition, we cannot see telomeric sequences (5'-TTTAGGG-3') at both and single ends of the chromosomes. It is not necessary to cite Fig. 1b here.

Response: Sorry for the confusion. There are two chromosomes with telomeric sequences at both ends and seven chromosomes with telomeric sequences at only one end. Fig. 1b shows detailed information of the presence/absence of telomeric sequences in each chromosome. We have revised the sentence to make it clearer in the revised manuscript (Lines 78-79).

Line 289, known desaturases and elongases? Elongase should be mentioned here.

Response: Done, thanks!

Reviewer #3 (Remarks to the Author):

Lou and colleagues successfully assembled *T. grandis* with such large genomes and elucidated seed development and gymnosperm-specific sciadonic acid biosynthesis through bioinformatics analysis. It is also very important to pay attention to the DNA methylation of large genomes. I was very interested to read this manuscript, but I still have some major concerns need to be clarified.

Response: We thank the reviewer for the positive comments.

Major concerns:

1. In lines 132 to 136, I was confused about the analysis in this part, especially the collinearity analysis couldn't tell me that *T. grandis* experienced two WGD events after the differentiation from *G. montanum*. Because the collinearity analysis failed to show 1:2(*G. montanum*: *G. biloba*) and 1:4 (*G. montanum*: *T.G. randis*) (Figure 1e). in addition, details of supplementary5 are also vague, hoping to elaborate on their views in more details.

Response: We are sorry for the confusion. Our gene tree-based analysis suggests four WGD events that had possibly occurred during the evolution of focal seed plants (Supplementary Fig. 4). Among these WGDs, the zeta and omega WGDs have also been previously reported in many studies; thus, we believe they are highly reliable. The newly discovered WGD (hereafter WGD3), shared by *G. biloba*, *P. tabuliformis*, *S. giganteum*, *T. grandis* and *T. wallichiana*, is supported not only by gene-tree based inference (Supplementary Fig. 4), but also by *Ks* analysis (Fig. 1d) and

collinear blocks (Fig. 1e). However, the potential most recent WGD (hereafter WGD4), shared by *S. giganteum*, *T. grandis* and *T. wallichiana* (Supplementary Fig. 4), was identified exclusively based on gene tree inference, and we could not find evidence from *Ks* analysis or synteny blocks (e.g., 1:4 ratio between *G. montanum* and *T. grandis*). It should be noted that one study (Li et al., *Sci. Adv.* 2015, 1:e1501084) that was cited in our original manuscript identified a WGD positionally similar to WGD4; however, it was also based only on gene tree analysis, and therefore, could be confounded by other factors such as gene duplications. In summary, our results suggest that *T. grandis* might have experienced only one ancient WGD (WGD3) event after divergence from the ancestor sharing with *G. montanum*.

We have modified the main text (Lines 124-125) and Supplementary Fig. 4 to avoid confusion and revised the legend of Supplementary Fig. 5 to make it clearer.

2. What percentage of genes have TE insertions, and whether these insertions occurred at intron or exon, or exon and intron coexist, their methylation levels, and whether there is an effect on host gene expression, and what effect?

Response: About 43% of *T. grandis* genes intersect with TEs, and 98.3% of them have TEs exclusively in introns, whereas the remaining genes have TEs either exclusively in exons (0.6%) or in both introns and exons (1.1%). We found that TE-inserted genes have higher expression level than non-TE genes, and larger TE insertion (e.g., >10kb) leads to higher gene expression than shorter TE insertion. This finding is consistent with that found in Chinese pine (Niu et al., *Cell*, 2022, 185:204-217). Moreover, we found a higher CG methylation level in TE-inserted genes than non-TE genes, and also in genes with larger TE insertions than those with shorter ones. In angiosperms, CG DNA methylation is enriched in the transcribed regions of many constitutively expressed genes, called gene body methylation (gbM). The evolution of gbM is hypothesized to be a byproduct of DNA methylation silencing of transposable elements (TEs) and other repetitive sequences in close proximity of genes (Bewick and Schmitz, *Curr Opin Plant Biol.* 2017, 36:103-110). Our data on the positive correlation of CG methylation, gene expression and TE insertion may suggest a mechanism similar to gbM in flowering plants that regulates DNA methylation and gene expression in TE-rich regions. We have added these new results in Supplementary Fig. 15 and included some of these points into the revised manuscript (Lines 318-322).

3. Figure 5d shows the methylation and insertion of intact LTRs. How does this relate to line 390-391.

Response: We are sorry for the confusion. This sentence has been rewritten in the revised manuscript (Line 320).

4. For DNA methylation analysis, I did not see the summary of sequencing depth and alignment ratio, conversion rate etc. I think the quality of methylation data is very important for the subsequent analysis. We know that it is very challenging to perform DNA methylation analysis of such a large genome by using second-generation sequencing of WGBS.

Response: Thanks for the understanding of the challenges for DNA methylation analysis of such a huge plant genome. We produced a large dataset (604-605 Gb, ~32×) for each of the samples and the conversion rate ranges between 99.737% to 99.749%. The average site coverage is 15.66-17.62×, and 77.52-82.83% of the genome has a site coverage of $\geq 10\times$ and 95.14-96.29% of the genome has a site coverage of $\geq 5\times$. The detailed statistics can now be found in Supplementary Table 12.

5. The authors failed to find abundant small 24nt RNAs in leaves, which may be because the sampling was not complete enough to infer functional differences of the RdDM pathway in gymnosperms.

Response: In this revised manuscript, we have generated sRNA sequences for seven additional samples, and the tissues used for sRNA sequencing now include leaf, female cone, male cone (two stages), kernel, stem, root tip, and aril. As shown in Supplementary Fig. 3, 21-nt sRNAs were indeed the most abundant in all these tissues, while productions of 22-nt and 24-nt sRNAs were tissue-specific. This pattern is similar to that observed in conifers (Nakamura et al., BMC Genomics, 2019, 20:997; Niu et al., Cell, 2022, 185:204-217) and *Welwitschia mirabilis* (Wan et al., Nat Commun., 2021, 12:4247), and implies the potential difference of sRNA processing between gymnosperms and angiosperms.

Minors:

1. The methods section, please add more methylation analysis methods and software parameters, including DMV, DMR recognition and so on

Response: Done (Lines 504-505). Thanks.

Reviewer #4 (Remarks to the Author):

This work details a high-quality chromosome-scale genome assembly that is an important and useful addition to the available genomics resources for conifers. The methods applied appear to be appropriate but there are additional methods details and summary analyses that I would like to see to establish further confidence in the assembly accuracy, in particular scaffolding results and haplotype purging, and annotation completeness. I appreciate that those details not being included is likely due to word limitations for the article type, but I do think a more extensive and complete methods description in the form of a supplementary note would be useful. One aspect that I find surprising is that the ab initio gene predictions tools are all described as working out of the box. This has not at all been my own experience with conifer genome, so I am curious to know if there are some details and filtering steps that were required that are not reflected in the current methods description.

Response: Thanks for the comments and please check our responses below for details.

It would also have been interesting to have some comparison to, and comment on, the differences to the results recently reported for the Chinese pine genome. There is a large difference in the reported number of annotated protein coding genes as well as noticeable differences in BUSCO scores. As these differences can influence the results of analyses of gene families and evidence for WGD events. Similarly, incomplete haplotig purging and influence such analyses. I was also surprised by the difference reported here for the solo to complete LTR ratio as this is in contrast to previous reports in conifers. Some brief comment on this in the conclusions would be interesting to include.

Response: The Chinese pine genome is reported to encode 80,495 predicted genes and 144,584 transcripts. Based on our evaluation, 49,991 gene models of Chinese pine were derived from transcriptome data, while 30,503 genes were from the MAKER pipeline (the remaining one was

manually identified). We speculate that quite a few of these genes (or transcripts) may not be authentic protein-coding genes as evidenced by incomplete CDS or pre-maturation found in these sequences and that only 58,214 genes (72.3%) have orthologues in other plant species (compared to 95.3% of the predicted *T. grandis* genes). BUSCO values are significantly different when assessed using proteins (84%) or the genome (44.5%) for Chinese pine. The authors reasoned that this was due to the failure of detection of super long genes with multiple introns. However, this was not the case in *T. grandis*, as the BUSCO assessment was similar for both proteins and the genome. The quality of genome annotation and haplotig purging are indeed the potential confounding factors for *Ks*-based WGD inference, but this impact is limited and mostly causes some extent of fake signal of very recent WGDs, which have rarely been reported in gymnosperms. As far as solo:intact LTR-RT ratio, we found that this ratio is large in many gymnosperms, including *T. grandis* (ratio=4.3), *Taxus wallichiana* (5.5; Chen et al., Mol Plant, 2021, 14:1199-1209), *Ginkgo biloba* (4.26), *Welwitschia mirabilis* (3.87), and *Gnetum montanum* (2.07; Wan et al., Nat Commun., 2021, 12:4247), which is highly different from that found in conifers, such as *Picea abies* (0.11; Nystedt et al., Nature, 2013,497:579-584) and *Pinus tabuliformis* (0.14-0.20; Niu et al., Cell, 2022, 185:204-217.e14). Genomes of gymnosperms are generally large, mainly contributed by the expansion of LTR-RTs. In contrast to many small-genome angiosperms, whose LTR-RTs are more recent (<4 mya), gymnosperms accumulate relatively ancient LTR-RTs (10-30 mya), with the exception of *P. tabuliformis*, which also has a recent burst of LTR-RTs (<6 mya). Based on these facts, we hypothesize that continuous elimination without remarkable expansion of LTR-RTs may have contributed to the high solo:intact LTR ratios in at least some gymnosperms. According to the reviewer's suggestion, we have included some of these points into the revised manuscript (Lines 101-106).

L34 The sentence should be in the past tense – so marked not marks.

Response: Done (Line 34).

L36 'diverged in the'

Response: Done (Line 36).

L37 earth should be Earth

Response: Done (Line 37).

L38 I do not understand what is meant by this sentence as it does not link to anything previously written. Apparent based on what?

Response: The sentence has been revised (Line 38).

L42 They are not nuts, they are seeds.

Response: We have changed it to "seeds" (Line 42).

L57 It is largely due to their large genomes sizes and high repeat content, not heterozygosity (there are numerous sequenced heterozygous angiosperms, for example).

Response: The sentence has been revised (Line 59-60).

L60 Anchored how?

Response: It was based on Hi-C data (Line 74-76). We have revised this entire paragraph (Line 55-66).

L71 About 96X and 23.2X (the decimal is not needed when using the qualifier ‘about’).

Response: We have removed “about” in the revised manuscript.

L76 A citation is needed for the 101 bp centromeric repeat.

Response: The centromeric repeat was identified in this study. We have included the method and relevant citations in the revised manuscript (Lines 428-429).

L82 It would be good to state the number of these that are complete and that are not duplicated.

Response: The numbers are detailed in Supplementary Table 3.

L108 Comparing to other gymnosperms with low contiguity assemblies is not appropriate. Intron sizes in all conifers appear to be similar (e.g. in Chinese pine, which is probably the other highest quality conifer assembly currently available).

Response: This sentence has been removed.

L112 This observation is in contrast to the results that have been reported for Norway spruce. Do the authors have an explanation for these contrasting results? A stronger force for removal runs counter to the large genome size of *T. grandis*.

Response: Please check our response above (question #2).

L117 It should be noted here that previous observation in Norway spruce identified that 24 nt sRNA expression is more dynamic and tissue dependent with results dependent on that sample assayed.

Response: In the revised manuscript, we have generated sRNA sequences for seven additional samples, and the tissues used for sRNA sequencing now include leaf, female cone, male cone (two stages), kernel, stem, root tip, and aril. As shown in Supplementary Fig. 3, 21-nt sRNAs were indeed the most abundant in all these tissues, while productions of 22-nt and 24-nt sRNAs were tissue-specific. This pattern is similar to that observed in conifers (Nakamura et al., BMC Genomics, 2019, 20:997; Niu et al., Cell, 2022, 185:204-217) and *Welwitschia mirabilis* (Wan et al., Nat Commun., 2021, 12:4247), and implies the potential difference of sRNA processing between gymnosperms and angiosperms.

L124 The spelling of paralog and orthologs is inconsistent here compared to the legend of Figure 1.

Response: Corrected.

L132 Are the authors confident that all genome assemblies and annotations used for this analysis are of sufficient (and comparable) quality to have confidence in these results? A number of factors, including the efficiency of haplotype purging, could influence the result.

Response: We agree with the reviewer that inefficient purging of haplotigs might potentially influence tree-based inference of WGDs, as the presence of additional alleles may lead to fake signal of gene duplication. However, allelic genes are highly similar, and therefore could be easily removed by setting a cutoff for sequence identity of paralogues, as what we did in our previous

study (Jiao et al., Cell 2020, 181:1097-1111.e12.). In this study, we have carefully examined the annotation of each species by removing isoforms and low-quality genes (e.g., incomplete CDS) prior to the analysis. Most importantly, WGDs identified in this study are ancient, which are unlikely influenced by the quality of assembly and annotations, as these factors, if present, may somehow cause difficulty in the determination of very recent WGDs.

L149 This sentence needs to point to where the result is reported, e.g. a table or figure (or a citation).
Response: Done (Line 145).

L159 As gymnosperms do not have true flowers this subheading is maybe not an ideal choice of wording

Response: We have changed it to “Reproductive organ genes” (Line 156).

L161 Homolog spelling

Response: We have made it consistent throughout the revised manuscript.

L256 VND, NST and SND are discussed in the NorWood wood development publication profiling gene expression during wood development in Norway spruce. These results would perhaps benefit from comparison to this previous report.

Response: Thanks for the suggestion. The findings of NorWood paper have been discussed in the revised manuscript (Lines 235-238).

L261 As *T. grandis* does not produce Paclitaxel I see little value in this section – although there is nothing wrong with the results reported here.

Response: This short paragraph has been removed. Please check our response to Reviewer #2 on the same issue.

L294 This result does not show that the genes are required in *T. grandis* as this would require a knock-out. It shows that they are capable of this.

Response: Thanks. The sentence has been revised according to the reviewer’s suggestion (Line 262).

Was there correspondence between the DMVs/DMRs and 21 nt sRNAs? What was the methylation status of repeats in the genome and were those consistent throughout seed development?

Response: Yes, we observed an enrichment of 21-nt sRNAs in DMVs/DMRs and this pattern was consistent for all tissues used from sRNA sequencing (see Fig. R1 below for examples). TEs were highly methylated in the *T. grandis* genome. In our original manuscript, we showed methylation status for LTR-RTs (Fig. 5d), which were the major group of TEs in the genome, but other TEs (e.g., DNA transposons) were also highly methylated. Moreover, we found that CG and CHG methylation of LTR-RTs were relatively conserved during seed development, but CHH methylation showed remarkable difference depending on developmental stages (see Fig. R2 below). The mechanisms underlying such variation are unclear, but this would definitely be an interesting question for our future studies.

Fig. R1. Size distribution of sRNA in DMRs and DMVs

Fig. R2. Methylation of LTR-RTs.

Conclusion 3 is not extensively supported by analyses presented in this paper. Similarly, while conclusion 2 is in line with generalizations, this is also not examined in detail with only gross-scale patterns reported here. For example, there is no examination of the relationship between methylation status and whether LTRs are present as complete or solo elements.

Response: The original conclusion 2 has been removed and conclusion 3 has been rewritten according to the reviewer's suggestions (Lines 357-361)

For conclusion 6 to be more fully evaluated would also require methylation data from non seed samples. It could be that these low methylation regions and more generally low methylation comply because they contain genes. There is no statistical support presented for these results.

Response: This conclusion was based on the statistical analysis of GO term enrichment as shown in Fig. 5e. The biological processes we mentioned here were those among the most significantly enriched ones. We believe that these results are new and relevant, and deserve a short summary in

the conclusion as this highlights our important findings on the potential epigenetic regulation of seed activities in *T. grandis*.

L437 I do not follow how this genome specifically fills a gap when other genomes for gymnosperms and conifers are already available. There are some compelling results in this paper specific to *T. grandis* but few new or novel results more generally.

Response: This sentence has been revised according to the reviewer's suggestion (Lines 369-370).

The conclusion would benefit from comparison to results from existing conifer genome publications, including work examine methylation patterns, differential expression and sRNA expression. This work is not performed in a void of available information, yet the conclusion section does not reflect this.

Response: The conclusion has been revised according to the reviewer's suggestions.

L444 The DNeasy kit does not extract truly HMW DNA and would not normally be recommended for use with PacBio long reads (although HiFi reads are admittedly not super long).

Response: We are sorry for the oversight here. HMW DNA was extracted using 2% CTAB, and we have corrected this in the revised manuscript (Line 376).

L460 Was RNA integrity assessed with e.g. a Bioanalyzer?

Response: Yes, it was our standard protocol to examine RNA integrity using Bioanalyzer 2100. RNA samples with a RIN score ≥ 8 were accepted for downstream experiments. We have clarified this in the revised manuscript (Line 393).

L461 How was rRNA depleted and what primers were used for cDNA synthesis?

Response: We used Oligo (dT) magnetic beads to purify mRNA, which was then used for library construction with the NEBNext Ultra II RNA Library Prep Kit for Illumina (NEB). Primers for cDNA synthesis were provided with the kit. This sentence has been revised (Line 393).

L463 Were the data strand specific or not?

Response: They were not strand specific. We have made it clear in the revised manuscript (Line 395).

L473 What parameters were set for Purge Haplotigs and how was the success of purging determined?

Response: The parameters for Purge Haplotigs have been detailed in the method (Lines 407-408). This software has been widely used in genome assembly papers and is proven to be efficient in purging haplotigs. We used stringent threshold (i.e., '-a 55') to remove redundant alleles while the non-redundant genetic elements were well preserved in the genome assembly as evidenced by the high mapping ratios of DNA and RNA sequencing reads.

L474 Were PCR duplicates removed (if so, how?) and was library complexity assessed?

Response: The Hi-C data were processed with the HiCUP pipeline (<https://www.bioinformatics.babraham.ac.uk/projects/hicup/>) to identify valid read pairs and alignments. PCR duplicates (5.22%) were removed, and the library complexity was assessed during the implementation of the pipeline. In this study, 56.94% of Hi-C reads were uniquely

mapped and 71.65% of them were valid pairs. We have detailed this in the revised methods (Lines 410-411).

L477 Assessed how?

Response: Illumina reads were mapped to the genome to assess the completeness of the assembly. We have replaced the “Quality” with “Completeness” in the revised manuscript (Line 413).

L481 What knowledge does this refer to?

Response: It means homology-based approach. We have revised it in the manuscript (Line 417).

L496 These tools all need parameter settings to be defined. Some also require a gold standard set of genes to be used for model training. How was this performed? These tools are known to perform poorly in conifer genomes due to the presence of long introns so how was this overcome? It is also very common that these *ab initio* methods will identify some additional repeats and pseudogenes, so was, or how was, this assessed? Were any filters applied for e.g. minimum gene or CDS length?

Response: Thanks for the comments. The parameters for gene predictors have been detailed in the methods (Lines 441-449). The *ab initio* predictors were trained with complete gene structures produced by the PASA pipeline which relies on transcriptome data of different tissues. We were aware of the potential issues of popular *ab initio* predictors for large genomes; therefore, gene structures derived from transcriptome data were preferred during the EVIDENCEModeler integration [weight score: PASA (based on transcriptome), 100; GeneWise (based on protein homology), 20; Cufflinks (based on transcriptome), 20; AUGUSTUS, 5; other *ab initio* predictors, 1]. We understand that there is still a need of the community effort to improve the performance of gene predictors for large genomes, but our pipeline provides a reasonable set of gene models that satisfy evolutionary and biological studies. Similar approaches have been used for gene predictions of other large genomes, including bunching onion (11.27 Gb; Liao et al., Nature Communications, 2022, 13:6690) and African lungfish (40Gb, Wang et al., Cell, 2021, 184:1362-1376). By the way, gene models in this study were not filtered with a length cutoff.

L507 Was BUSCO applied to the set of predicted proteins or to the genome, or both?

Response: BUSCO assessment was applied to both genome and predicted proteins, and the statistics were similar (Supplementary Table 3).

L543 Describe the sample used for this DNA extraction.

Response: Done (Lines 486-487).

How many genes were derived from the different evidence types and how many were commonly supported? The information presented about this is very minimal.

Response: Among the 47,089 predicted genes, 44,863 were supported by homology-based evidence from public databases, and 33,994 were supported by our RNA-Seq dataset. There were 32,519 genes supported by both protein homology and RNA-Seq evidence, and 46,338 genes supported by at least one of the two evidences. We have now made this clear in Supplementary Table 2.

The methods do not detail expression analysis of the RNASeq data.

Response: Methods for RNA-Seq data analysis have been added in the revised manuscript (Lines 507-515).

Figure 1. Why are different track types used for the Gypsy and Copia elements?

Response: In flowering plants, Gypsy and Copia retrotransposons are known to have different distribution pattern on chromosomes, i.e., Gypsy retrotransposons are concentrated near centromeres while Copia retrotransposons are distributed across the chromosomes (Bennetzen and Wang, *Annu Rev Plant Biol.*, 2014,65:505-530). So here the Gypsy and Copia retrotransposons were plotted separately to show their chromosomal distributions.

Figure S1 The figure is quite low resolution so it is hard to view in detail but there are some features in the contact map that appear to be potential miss-assemblies. How was this assessed and was any manual curation performed after the initial Hi-C scaffolding?

Response: The Hi-C scaffolding was manually curated using Juicebox based on the intensity and pattern of interaction maps, which has been clarified in the revised manuscript (Lines 412-413). To make a better visualization, we have provided Hi-C map for each chromosome in Supplementary Fig. 2.

Figure S3 While 24nt sRNA are relatively low abundance it would be interesting to know which genomic features these aligned to. Similarly, is there a population of 21nt sRNAs that are repeat associated and that correspond to methylation regions?

Response: We examined the genomic features to which the 21- and 24-nt sRNAs align. As shown in Fig. R3 below, 24-nt sRNAs were predominantly produced from TE regions, while 21-nt sRNAs were generated mostly from genes but also with a certain proportion from TEs. This pattern is consistent with the findings in conifers (Nakamura et al., *BMC Genomics*, 2019,20:997). Moreover, the 21-nt sRNAs were also found to be enriched in DMRs (Fig. R1 above), suggesting a potential link between 21-nt sRNAs and the methylation of the genome. We note that these points are very interesting, but they are beyond the scope of this study and may require more data and rigorous analysis to verify; therefore, we did not include these points in the revised manuscript.

Fig. R3 Proportion of 21- and 24-nt sRNA from gene and TE regions.

Figure S17 This is not a correlation

Response: These figures show correlation coefficients of methylation profiles among different samples. We have made this clear in the corresponding figure.

Reviewers' Comments:

Reviewer #1:

Remarks to the Author:

Review of manuscript NCOMMS-22-43912-T

I reviewed the previous version of the manuscript and there is still something that is not clear to me in relation to the frequency of small RNAs.

My question was:

Also, the evidence for the observation that 24-nt are less abundant than 21-nt comes from the analysis of needle tissue (suppl figure 3) when it has been reported that 24-nt are more abundant in developing seedlings and basal meristems in other gymnosperm species (such as *Welwitschia* and others). More evidence from other tissues is required to confirm the (potentially different) small RNAs patterns in gymnosperms.

Response: Thanks. According to the reviewer's suggestion, we have generated sRNA sequences for seven additional samples, and the tissues used for sRNA sequencing now include leaf, female cone, male cone (two stages), kernel, stem, root tip, and aril. As shown in Supplementary Fig. 3, 21-nt sRNAs were indeed the most abundant in all these tissues, while productions of 22-nt and 24-nt sRNAs were tissue-specific. This pattern is similar to that observed in conifers (Nakamura et al., *BMC Genomics*, 2019, 20:997; Niu et al., *Cell*, 2022, 185:204-217) and *Welwitschia mirabilis* (Wan et al., *Nat Commun.*, 2021, 12:4247), and implies the potential difference of sRNA processing between gymnosperms and angiosperms.

I will start by thanking the authors for including the new data from other tissues. I must say, however, that the new data does not provide evidence to my original question of whether the lower abundance of 24-nt in this study might be an underestimation due to the lack of sampling in developing seedlings and basal meristems. I assume that none of the new tissues correspond to these tissue types, therefore this question remains unanswered. I would suggest explaining the limitations of the results of this work in lines 107-113.

Minor comments:

- Line 32- The last sentence of the abstract is not clear, please rephrase.
- Line 40- A citation is missing.
- Line 74- Please include N50 for contigs and scaffolds. Is the contiguity similar to other chromosome-scale gymnosperm genomes?
- Line 78-79- *T. grandis* has a lower repetitive content than most gymnosperms sequenced to date, could this be an underestimation due to the lack of telomeres or other parts of the genome?
- Lines 104-105- The sentence "...continuous elimination without remarkable expansion of LTR-RTs.." seems speculative and not so clear, please rephrase.
- Line 112- It looks like some citations are missing based on the "Response to Reviewers" document (paragraph above).
- Line 376- CTAB does not produce HMW DNA.
- Line 386- company for Hi-C?
- Line 488- Please add details of the protocol for sonication.

Reviewer #2:

Remarks to the Author:

I have gone through the revised manuscript and authors' response. All my concerns or suggestions have been well addressed. The figures are nicely organized, and the writing is good.

Reviewer #3:

Remarks to the Author:

The authors have replied to most of my queries and have modified them accordingly. I still have some questions about the part of methylation analysis, especially the statement of gbM. gbM is defined in angiosperms as high CG methylation level, whereas low non-CG methylation level at gene body regions. This is different from the DNA methylation distribution in *T. grandis*. We could observe very high CHG methylation at gene body region, partly due to TE insertions at gene region. So the statement about gbM still needs to be modified.

In addition, the genome of *T. grandis* has a high integrity compared to other gymnosperms, but there are still some functionally important genes not identified in the identification of methylation pathway-related genes in supplementary table 5. And I noticed that the corresponding genes in Arabidopsis were also not found, please double check. It would be better to do sequence alignment or phylogenetic tree construction for several important genes, such as CMT3. Especially CMT3 is crucial in the formation of gbM.

Finally, I understand the difficulty of analyzing methylation in such a large genome using WGBS, but given the completeness of the genome assembly and the high depth of WGBS sequencing, I have reason to consider that the methylation analysis in this paper will go deeper than the results of known gymnosperms, which is important for our understanding of DNA methylation in gymnosperms

Reviewer #4:

Remarks to the Author:

The authors have addressed both my comments and those of the other reviewers and have revised the manuscript accordingly. It is great to see the additional of the expanded set of sRNA results. I think the authors for their responses. The revised manuscript has addressed my original points and I now consider the work suitable for publication.

Reviewer #1

Review of manuscript NCOMMS-22-43912-T

I reviewed the previous version of the manuscript and there is still something that is not clear to me in relation to the frequency of small RNAs.

My question was:

Also, the evidence for the observation that 24-nt are less abundant than 21-nt comes from the analysis of needle tissue (suppl figure 3) when it has been reported that 24-nt are more abundant in developing seedlings and basal meristems in other gymnosperm species (such as *Welwitschia* and others). More evidence from other tissues is required to confirm the (potentially different) small RNAs patterns in gymnosperms.

Response: Thanks. According to the reviewer's suggestion, we have generated sRNA sequences for seven additional samples, and the tissues used for sRNA sequencing now include leaf, female cone, male cone (two stages), kernel, stem, root tip, and aril. As shown in Supplementary Fig. 3, 21-nt sRNAs were indeed the most abundant in all these tissues, while productions of 22-nt and 24-nt sRNAs were tissue-specific. This pattern is similar to that observed in conifers (Nakamura et al., *BMC Genomics*, 2019, 20:997; Niu et al., *Cell*, 2022, 185:204-217) and *Welwitschia mirabilis* (Wan et al., *Nat Commun.*, 2021, 12:4247), and implies the potential difference of sRNA processing between gymnosperms and angiosperms.

I will start by thanking the authors for including the new data from other tissues. I must say, however, that the new data does not provide evidence to my original question of whether the lower abundance of 24-nt in this study might be an underestimation due to the lack of sampling in developing seedlings and basal meristems. I assume that none of the new tissues correspond to these tissue types, therefore this question remains unanswered. I would suggest explaining the limitations of the results of this work in lines 107-113.

Response: It is worth pointing out that after carefully checking Fig. S5a of the *Welwitschia mirabilis* genome paper (<https://www.nature.com/articles/s41467-021-24528-4>) and Fig. S2.10 of the Norway spruce genome paper (<https://www.nature.com/articles/nature12211>; mentioned previously by Reviewer 4), we found out that 21-nt sRNAs were indeed more abundant than 24-nt sRNAs in all samples reported in these two studies, including meristem tissues. Nonetheless, according to reviewer's suggestion, we have mentioned the potential limitation of the sampling in the revised manuscript (Line 114-115).

Minor comments:

- Line 32- The last sentence of the abstract is not clear, please rephrase.

Response: Done. Thanks.

- Line 40- A citation is missing.

Response: Two related references have been added here.

- Line 74- Please include N50 for contigs and scaffolds. Is the contiguity similar to other chromosome-scale gymnosperm genomes?

Response: The contig N50 (2.82 Mb) was already mentioned in the original manuscript (Line 74). We did not include the scaffold N50 (1.824 Gb) in the manuscript because this

value is not that meaningful for chromosome-level scaffolds as it largely depends on the size of individual chromosome and does not really reflect the quality of the assembly (chromosome sizes of the *Torreya grandis* assembly are provided in Table S2). Despite the much larger genome size of *T. grandis* (19 Gb), the contiguity of its genome assembly at the contig level is comparable to or better than that of most other reported gymnosperm genome assemblies, e.g., *Taxus chinensis* (genome size: 10.23Gb; contig N50: 2.44 Mb), *Welwitschia mirabilis* (genome size: 6.86 Gb; contig N50: 1.48 Mb), *Gnetum montanum* (genome size: 4.07 Gb; contig N50: 25.02 kb), *Ginkgo biloba* (genome size: 9.87Gb; contig N50: 1.58 Mb), and *Pinus tabuliformis* (genome size: 25.4 Gb; contig N50: 2.6 Mb).

- Line 78-79- *T. grandis* has a lower repetitive content than most gymnosperms sequenced to date, could this be an underestimation due to the lack of telomeres or other parts of the genome?

Response: We agree with the reviewer that incomplete assembly of telomeres and other repetitive regions in the genome could be a factor for the relatively low repeat content of the *T. grandis* genome. It is worth noting that this should also apply to many, if not all, other gymnosperm genome assemblies, which also lack some telomeres and highly repetitive regions. To avoid possible confusion, we have changed the sentence “The *T. grandis* genome harbored 11.4 Gb (59.8%) of repetitive sequences” to “The *T. grandis* genome **assembly** harbored 11.4 Gb (59.8%) of repetitive sequences” (Line 87).

- Lines 104-105- The sentence “...continuous elimination without remarkable expansion of LTR-RTs..” seems speculative and not so clear, please rephrase.

Response: Done (Line 105-106). Thanks.

- Line 112- It looks like some citations are missing based on the “Response to Reviewers” document (paragraph above).

Response: Thanks. References have been added (Line 114).

- Line 376- CTAB does not produce HMW DNA.

Response: Thanks for pointing this out! We have removed the “high-molecular-weight” in the revised manuscript. DNA for HiFi sequencing was extracted using the CTAB method, which is true that it is not HMW DNA.

- Line 386- company for Hi-C?

Response: Methods for Hi-C were rephrased to include company and other information (Line 385-392).

- Line 488- Please add details of the protocol for sonication.

Response: Done (Line 493-494). Thanks.

Reviewer #2

I have gone through the revised manuscript and authors' response. All my concerns or suggestions have been well addressed. The figures are nicely organized, and the writing is good.

Response: Thanks!

Reviewer #3

The authors have replied to most of my queries and have modified them accordingly. I still have some questions about the part of methylation analysis, especially the statement of gbM. gbM is defined in angiosperms as high CG methylation level, whereas low non-CG methylation level at gene body regions. This is different from the DNA methylation distribution in *T. grandis*. We could observe very high CHG methylation at gene body region, partly due to TE insertions at gene region. So the statement about gbM still needs to be modified.

Response: Thanks for the suggestion! The very high CHG methylation in gene body (including exons and introns; Figure S14a) indeed indicates the insertion of TEs in introns. In the revised manuscript, we have included a new plot showing methylation levels only in exons (Figure S14b), which again shows the enrichment of CG and CHG methylation (a lower level) in exons of *T. grandis* genes, and this pattern is similar to that found in conifers (<https://www.nature.com/articles/nplants2015222>). According to reviewer's suggestion, we have modified the corresponding sentences in the revised manuscript (Line 313-318).

In addition, the genome of *T. grandis* has a high integrity compared to other gymnosperms, but there are still some functionally important genes not identified in the identification of methylation pathway-related genes in supplementary table 5. And I noticed that the corresponding genes in Arabidopsis were also not found, please double check. It would be better to do sequence alignment or phylogenetic tree construction for several important genes, such as CMT3. Especially CMT3 is crucial in the formation of gbM.

Response: Thanks. We have updated Supplementary Table 5 with careful examination of each gene. We note that the homologous genes in different species were indeed identified based on the phylogeny constructed for each orthologous group that contains the representative Arabidopsis gene. We have also included this information in the header of Supplementary Table 5.

Finally, I understand the difficulty of analyzing methylation in such a large genome using WGBS, but given the completeness of the genome assembly and the high depth of WGBS sequencing, I have reason to consider that the methylation analysis in this paper will go deeper than the results of known gymnosperms, which is important for our understanding of DNA methylation in gymnosperms.

Response: We thank the reviewer for the helpful and constructive comments on the methylation results, which have improved the quality of the manuscript. Moreover, we believe that our results on DNA methylation extend current knowledge on the conservation of DNA methylation in flowering and non-flowering plants and provide new insights into the epigenetic regulation of seed development in gymnosperms.

Reviewer #4

The authors have addressed both my comments and those of the other reviewers and have revised the manuscript accordingly. It is great to see the additional of the expanded set of

sRNA results. I thank the authors for their responses. The revised manuscript has addressed my original points and I now consider the work suitable for publication.

Response: Thanks!

Reviewers' Comments:

Reviewer #1:

Remarks to the Author:

The authors have responded to all the points raised by me and other reviewers. I believe the manuscript is in good shape and I have no further questions.

Reviewer #3:

Remarks to the Author:

The revised manuscript has addressed all my concerns and the analysis and results are well responded. I do not have any further questions.

Reviewer #1

The authors have responded to all the points raised by me and other reviewers. I believe the manuscript is in good shape and I have no further questions.

Response: Thanks.

Reviewer #3

The revised manuscript has addressed all my concerns and the analysis and results are well responded. I do not have any further questions.

Response: Thanks.